# Inhibition of amyloid beta toxicity in zebrafish with a chaperone-gold nanoparticle dual strategy

Ibrahim Javed [1,2], Guotao Peng [2], Yanting Xing[3], Tianyu Yu[2], Mei Zhao[2], Aleksandr Kakinen[1], Ava Faridi[1], Clare L. Parish[4], Feng Ding [3], Thomas P. Davis [1,5], Pu Chun Ke [1] & Sijie Lin [2]

Alzheimer's disease (AD) is the most prevalent form of neurodegenerative disorders, yet no major breakthroughs have been made in AD human trials and the disease remains a paramount challenge and a stigma in medicine. Here we eliminate the toxicity of amyloid beta (Aβ) in a facile, high-throughput zebrafish (*Danio rerio*) model using casein coated-gold nanoparticles (βCas AuNPs). βCas AuNPs in systemic circulation translocate across the blood brain barrier of zebrafish larvae and sequester intracerebral Aβ$_{42}$ and its elicited toxicity in a nonspecific, chaperone-like manner. This is evidenced by behavioral pathology, reactive oxygen species and neuronal dysfunction biomarkers assays, complemented by brain histology and inductively coupled plasma-mass spectroscopy. We further demonstrate the capacity of βCas AuNPs in recovering the mobility and cognitive function of adult zebrafish exposed to Aβ. This potent, safe-to-use, and easy-to-apply nanomedicine may find broad use for eradicating toxic amyloid proteins implicated in a range of human diseases.

[1] ARC Centre of Excellence in Convergent Bio-Nano Science and Technology, Monash Institute of Pharmaceutical Sciences, Monash University, 381 Royal Parade, Parkville, VIC 3052, Australia. [2] College of Environmental Science and Engineering, Biomedical Multidisciplinary Innovation Research Institute, Shanghai East Hospital, Shanghai Institute of Pollution Control and Ecological Security, Key Laboratory of Yangtze River Water Environment, Tongji University, 1239 Siping Road, Shanghai 200092, China. [3] Department of Physics and Astronomy, Clemson University, Clemson, SC 29634, USA. [4] The Florey Institute of Neuroscience and Mental Health, The University of Melbourne, 30 Royal Parade, Parkville, VIC 3052, Australia. [5] Australian Institute for Bioengineering and Nanotechnology, The University of Queensland, Brisbane, Qld 4072, Australia. Correspondence and requests for materials should be addressed to F.D. (email: fding@clemson.edu) or to T.P.D. (email: thomas.p.davis@monash.edu) or to P.C.K. (email: pu-chun.ke@monash.edu) or to S.L. (email: lin.sijie@tongji.edu.cn)

The aggregation of proteins into amyloid fibrils and plaques, under abnormal physiological conditions, is a phenomenon common to a range of human amyloid diseases including amyloid beta (Aβ) for Alzheimer's disease (AD), α-synuclein for Parkinson's disease (PD), and human islet amyloid polypeptide for type 2 diabetes (T2D)[1]. The amyloid hypothesis regards oligomers as the most toxic species[2], where protofibrils or oligomers of amyloid proteins are proposed to induce local inflammation, failed autophagy, and membrane perturbation that are responsible for the further loss of neuronal or pancreatic β-cells mass[3,4].

AD is a primary form chronic neurodegenerative disorder and a major cause of dementia, impairing 46 million people worldwide[5]. The pathological origin of AD is highly debatable, but is believed to be associated with a range of health, genetics, environmental and lifestyle factors, as well as inflammation[6–9]. The etiology of AD includes a number of events that precede Aβ plaque formation, such as autophagy or endosomal dysfunction[10], endoplasmic reticulum stress[11], oxidative stress or hypoxia, vasculature and mitochondrial dysfunction[12], and prior history of bacterial infections[13]. Aβ$_{42}$ is one of the two most abundant peptide species derived from amyloid precursor protein (APP) through proteolysis and, alongside tau, is strongly associated with the pathology of AD[14]. Despite much research over the past decades devoted to understanding the origin, diagnosis and prevention of AD, there is a glaring lack of success against Aβ amyloidosis marked by recent withdrawals of clinical trials with Eli Lilly, Pfizer, and Biogen[15–17]. This indicates failures in current anti-amyloid therapeutic approaches, compounded by a lack of suitable in vivo models for high throughput screening[18,19], further justifying the urgency for developing alternative strategies against AD.

Among the common strategies against amyloidosis, peptides, small molecules, monoclonal antibodies and, more recently, engineered nanoparticles, have shown various degrees of promise as inhibitors[20–27]. For in vivo applications, these inhibitors are designed to satisfy—partially or fully—the following criteria: minimal toxicity, good circulation/repeated dosing, good translocation efficacy across the blood brain barrier (BBB), as well as capabilities in targeting and further eliminating toxic oligomers, protofibrils, and fibrils of amyloid proteins. β casein (βCas), a whey protein, along with α$_{s1}$ casein, possesses a chaperone-like activity, similarly to small heat-shock proteins and extracellular clusterin. This activity of the caseins arises from the following: (1) a lack of tertiary structure and solvent-exposed hydrophobicity with well separated hydrophilic regions, (2) existence as heterogeneous oligomers, (3) dynamics and malleable protein regions, and (4) ability to bind with a wide range of partially folded proteins preventing their aggregation[28]. One factor that attributes to these properties is the presence of a high percentage of proline residues, i.e., 18% in the case of βCas, and no disulfide bonds that provide them with an open and flexible conformation[29]. The chaperone-like behavior of βCas and α$_{s1}$ caseins shields the amyloidogenic regions and naturally prevents the amyloidosis of α$_{s2}$ and κ-casein in mammary glands or milk while inhibiting the amyloidosis of insulin and Aβ$_{40}$ in vitro[30–32]. Structurally, monomeric caseins are mostly disordered, but tend to form micelles mediated by hydrophobic and electrostatic interactions[28].

Here, we devise a facile method of coating βCas onto gold nanoparticles (AuNPs). We systemically deliver the βCas AuNPs via intracardial administration to mitigate the toxicity of Aβ$_{42}$ induced in the brain of zebrafish larvae and adults (*Danio rerio*). βCas AuNPs sequester toxic Aβ$_{42}$ in the brain of zebrafish larvae and adults through a nonspecific, chaperone-like manner. No such mitigation is obtained with caseins alone, indicating the essential role of the AuNPs in delivering the protein. This demonstrates the inhibition potential of a chaperone protein integrated with a biocompatible nanomaterial against Alzheimer's-like symptoms. The established zebrafish model also opens the door to economically viable, high-throughput in vivo screening of emerging nanomedicines targeting a wide range of amyloid diseases.

## Results and discussion

**Scheme of study.** Different fractions of caseins, e.g., α$_{s1}$ and β, have a known potential for surface-assisted sequestration and colloidal inhibition of Aβ$_{40}$ and insulin amyloid formation[31,32]. Herein, βCas with an intrinsic chaperone-like activity[33] was coated on AuNPs by NaBH$_4$-assisted reduction of Au. Synthesis of the βCas AuNPs was optimized at room temperature to obtain ~5 nm in size for efficient BBB translocation while preserving the random coil structure of βCas that is required for its chaperone activity[30]. βCas AuNPs were then characterized for their inhibitory activity against Aβ$_{42}$ (abbreviated as Aβ from hereon) fibrillization in vitro. For in vivo translation, an Aβ toxicity model was developed in zebrafish larvae by cerebroventricular injection of Aβ. The biodistribution and translocation of βCas AuNPs across the larval zebrafish BBB were then determined after introducing the nanoparticles into the bloodstream via intracardiac injection. Finally, Aβ and βCas AuNPs were co-administered via cerebroventricular and intracardiac injections into zebrafish larvae, and alleviations of Aβ-induced behavioral symptoms were quantified. In addition, Aβ-induced behavioral pathology and cognitive dysfunction in adult zebrafish were rescued by βCas AuNPs, further implicating the chaperone potential of βCas AuNPs against Aβ toxicity in vivo.

**In vitro interaction of βCas AuNPs and Aβ.** Aβ was fibrillized in vitro from random coils to β-sheet rich amyloid fibrils within 48 h at 37 °C. A thioflavin T (ThT) assay was used to study the fibrillization kinetics (Fig. 1a), while transmission electron microscopy (TEM) (Fig. 1b) and circular dichroism (CD) spectroscopy were employed to investigate fibril formation and secondary structural transitions of Aβ (Fig. 1c, d). βCas AuNPs (Fig. 1e) often clustered together after binding with Aβ (Fig. 1f) and prevented β-sheet formation of the peptide (Fig. 1a). The presence of Aβ coronae on βCas AuNPs was evident from TEM imaging (Fig. 1f inset). The secondary structure of βCas AuNPs was predominantly random coils that transitioned into α helices in Aβ-βCas AuNPs complex (Fig. 1g, d). βCas formed micelles of ~100 nm in size in the aqueous medium (Fig. 1h). Upon incubation with Aβ, βCas (in the absence of the AuNPs) induced an early onset of fibrillization as revealed by the ThT assay (Fig. 1a), which can be attributed to the fast nucleation of Aβ promoted by βCas micelles in vicinity[34]. However, Aβ fibrillization was not inhibited and random aggregates of the peptide were observed under TEM (Fig. 1i). CD spectroscopy indicated that the α-helix rich structure of βCas was converted to β-sheets in βCas + Aβ aggregates due to Aβ fibrillization (Fig. 1j, d). The hydrodynamic diameters of βCas AuNPs and βCas were increased from 7.5 ± 2.6 and 156.3 ± 34.4 nm to 39.3 ± 5.4 and 496.1 ± 114 nm (n = 3), respectively (Fig. 1k and Supplementary Table 1). The zeta potential of βCas AuNPs was markedly elevated from −11.7 ± 1.8 to −33.7 ± 2.1 mV (n = 3), indicating adsorption of anionic Aβ onto the surfaces of βCas AuNPs (Supplementary Table 1). Clusterization of βCas AuNPs was confirmed by hyperspectral imaging (HSI), where the surface plasmon resonance (SPR) of βCas AuNPs was red shifted from 490 ± 21 to 601 ± 24 nm (n = 3) upon aggregation and light illumination (Supplementary Fig. 1A–C). As oligomers/protofibrils are the main toxic species

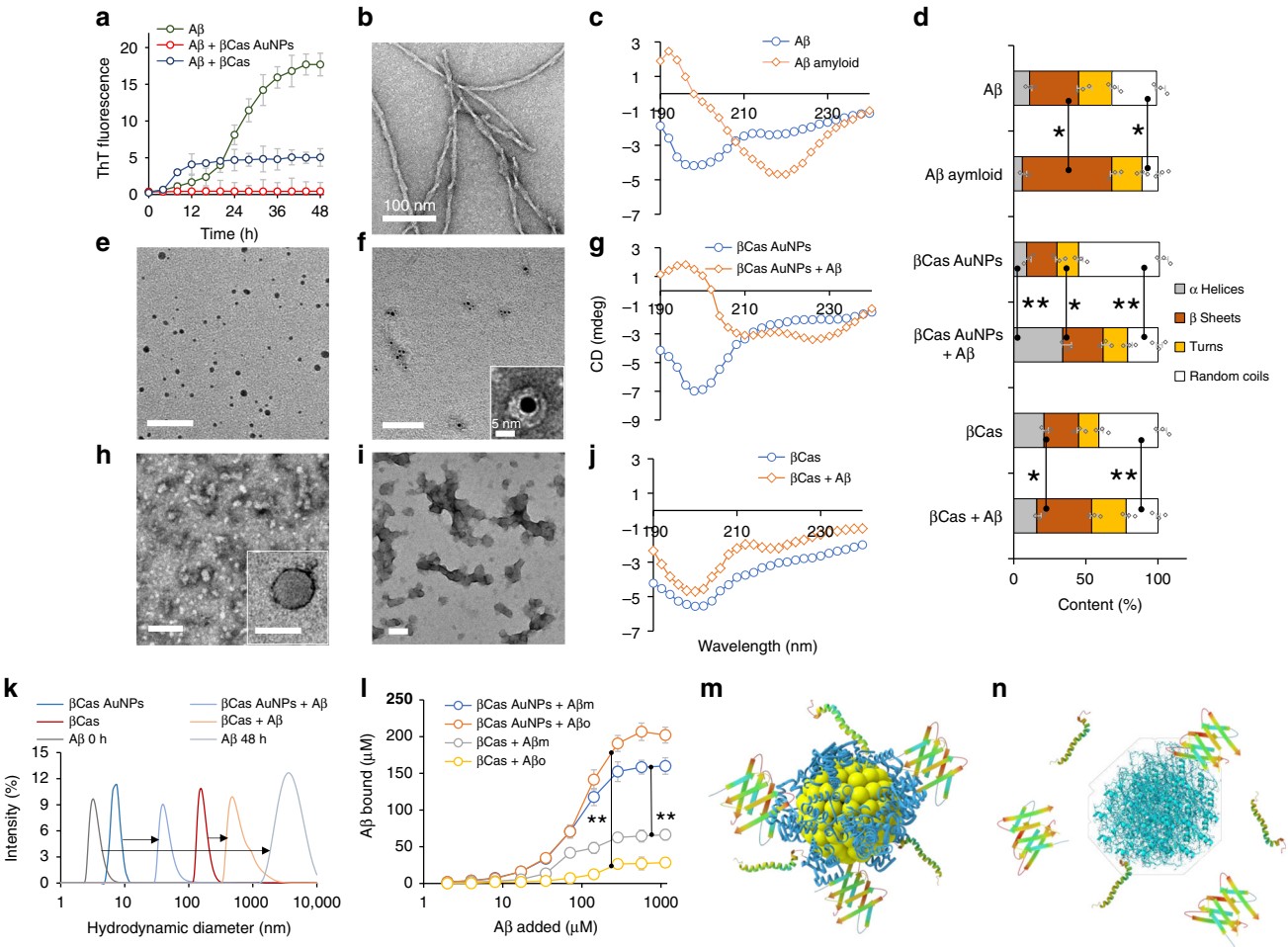

**Fig. 1** In vitro inhibitory interactions between βCas AuNPs and βCas with Aβ. **a** ThT assay of Aβ alone (50 μM) and in the presence of βCas AuNPs (equivalent to 6.25 μM of βCas) and βCas (6.25 μM) ($n = 4$). βCas AuNPs completely inhibited while βCas decreased ($p < 0.005$) the lag time and plateau ThT fluorescence of Aβ fibrillization. **b** TEM image of fibrillized Aβ. **c** CD spectra of Aβ (100 μM) before and after fibrillization indicate conformation change from random coils (198 nm peak) to β-sheets (220 nm peak). **d** Secondary structure of Aβ (100 μM) fibrillized with and without βCas AuNPs or βCas ($n = 4$). TEM images of βCas AuNPs before (**e**) and after incubation with Aβ (**f**; inset shows Aβ corona on a βCas AuNP). **g** CD spectra of βCas AuNPs before and after incubation with Aβ. TEM images of βCas before (**h**) and after (**i**) incubation with Aβ. **j** Appearance of a negative peak at 218 nm in CD spectra of βCas + Aβ indicates limited Aβ fibrillization into β-sheets. After binding with Aβ, the α-helix contents of βCas AuNPs were decreased significantly ($p < 0.05$) from 56 to 21%; while in the case of βCas, α-helices decreased from 41 to 22% ($p < 0.005$) and β-sheets increased from 24 to 38% ($p < 0.05$). **k** Hydrodynamic radius of Aβ before and after fibrillization in the presence and absence of βCas AuNPs or βCas. **l** Quantification of binding capacity of βCas AuNPs or βCas with Aβm or Aβo ($n = 4$). βCas AuNPs (6.25 μM βCas equivalent) were able to bind up to 152.3 ± 13.4 and 190.7 ± 10.9 μM of Aβm and Aβo. βCas (6.25 μM) was only able to adsorb 62.4 ± 3.4 and 26.4 ± 4.6 μM of Aβm and Aβo, indicating a significant ($p < 0.005$) increase in Aβ binding capacity of βCas in the form of βCas AuNPs. **m, n** Enhanced binding of Aβm and Aβo with βCas promoted by the AuNP substrate. Scale bars in TEM images is 100 nm, while F inset is 5 nm. Error bars represent the standard deviation. Source data are provided as a Source Data file

of amyloid proteins[14], interactions of βCas AuNPs with Aβ monomers (Aβm) and Aβ oligomers (Aβo) were also examined. The in vitro binding between Aβo/m and βCas or βCas AuNPs was further quantified by a bicinchoninic acid assay (BCA) and thermogravimetric analysis (TGA). βCas or βCas AuNPs were incubated with different concentrations of Aβm or Aβo for 48 h and centrifuged to remove free Aβ. The centrifuged pellets containing Aβm or Aβo bound to βCas or βCas AuNPs were subjected to analysis (Supplementary Fig. 1D). The maximum binding capacity between Aβ and βCas (6.25 μM) was quantified to be 62 and 26 μM for Aβm and Aβo, respectively (Fig. 1l). However, when βCas AuNPs (containing 6.25 μM βCas) were exposed to Aβ, the maximum binding capacity was increased to 152 and 190 μM for Aβm and Aβo, respectively. Similar results were obtained with TGA, where no difference in the TGA curve was observed when the concentration of Aβ was increased beyond

0.3 and 0.06 mM for βCas AuNPs and βCas, respectively, suggesting binding saturations (Supplementary Fig. 1E). The differential binding of βCas and βCas AuNPs with Aβm/o is illustrated in Fig. 1m, n.

The high affinity of βCas AuNPs for Aβo can be attributed to the ability of βCas to bind with misfolded/molten globules of proteins[28]. To further investigate the differential binding of βCas AuNPs with Aβo and Aβm, we incubated βCas AuNPs with preformed Aβo and Aβm for 3 h and separated them from unbound Aβo/m via centrifugal washing. The UV-SPR spectra of βCas AuNPs were significantly suppressed upon incubation with Aβo as compared to Aβm (Supplementary Fig. 2A). Similarly, the fluorescence of neutral red-conjugated AuNPs (NR-βCas AuNPs) was suppressed when incubated with Aβo (Supplementary Fig. 2B). This indicates increased adsorption of Aβo than Aβm by βCas AuNPs, as further confirmed by TEM imaging of corona

formation on the nanoparticles (Supplementary Fig. 2C). Furthermore, CD results indicated similar secondary structural distributions of Aβo and Aβo-βCas AuNPs complex. Thus, the α helices in Aβo-βCas AuNPs complex can be attributed to the Aβo corona on βCas AuNPs (Supplementary Fig. 2D, E). Incubation of βCas AuNPs with Aβm did not present any difference in the secondary structure of βCas AuNPs. That, together with the UV-SPR, fluorescence and TEM results, confirmed the high affinity of βCas AuNPs for Aβo.

**DMD simulations of βCas binding with AuNP and Aβ monomer/oligomer**. To gain a molecular insight into the adsorption of βCas onto an AuNP surface (i.e., the formation of a βCas AuNP "corona") and the inhibition mechanism of βCas AuNPs against Aβ aggregation, discrete molecular dynamics (DMD) simulations—an accurate and rapid molecular dynamics algorithm widely used to study the structure and dynamics of large molecular systems[35,36]—were performed (Fig. 2). The binding of a βCas monomer with an AuNP (4 nm in diameter), an Aβ monomer and an Aβ oligomer were examined (Supplementary Methods), and the control simulations included an isolated βCas, an Aβ monomer, and an Aβ oligomer. We first computed secondary structure contents from equilibrium simulations (e.g., radius of gyration in Supplementary Fig. 3A and number of hydrogen bond in Supplementary Fig. 5A indicated simulations reaching steady states) and used them to estimate the expected CD spectra for different molecular systems (Fig. 2b). The predicted CD spectra agreed well with the experimental results (Fig. 1c, g, j, d) in terms of secondary structural changes. As expected, βCas was intrinsically disordered with unstructured coils as the dominant secondary structure. Upon binding the

AuNP, βCas exhibited increased coil and decreased helix and sheet contents both in silico (Fig. 2b inset) and in vitro (Fig. 1g, j, d). Analysis identified several specific binding sites of βCas for the AuNP, such as residues His65, Phe67, Lys122, Met124, and His159 (the upper panel in Fig. 2a). Based on clustering analysis of the structural ensemble from multiple independent DMD simulations (Supplementary Methods), representative binding structures of βCas monomers with the AuNP were obtained (Supplementary Fig. 3B), where individual βCas partially covered the AuNP. To form a monolayer protein corona, at least three βCas molecules were required to fully coat the AuNP surface (Fig. 2c, estimated by covering the NP surface with randomly selected centroid structures from top ten clusters shown in Supplementary Fig. 3B). When βCas bound to an Aβ monomer, the overall contents of ordered helices and sheets increased while coils decreased (Fig. 2b), in agreement with the experiments (Fig. 1c, j, d). Residues in βCas that had strong binding with the Aβ monomer did not overlap with those preferred to bind the AuNP (Fig. 2a). This result suggests that βCas AuNP could still bind to Aβ monomers, as illustrated by the βCas-AuNP complex where Aβ-binding residues were exposed (Fig. 2c, with the protein surface color-coded according to their binding probabilities with the Aβ monomer). Although we did not perform simulations for the binding of Aβ with βCas AuNP, due to the prohibitively large system, we expected similar trends of secondary structure changes as observed in the experiments (Fig. 1d). Representative structures of the binding complexes obtained from the simulations (Supplementary Fig. 4A) suggest that βCas could bind Aβ and form either β sheets or helices (Supplementary Fig. 4B), which in turn inhibited Aβ aggregation by sequestering Aβ in solution or capping Aβ fibrils from elongation. Moreover, simulations of βCas with a preformed cross-β Aβ oligomer

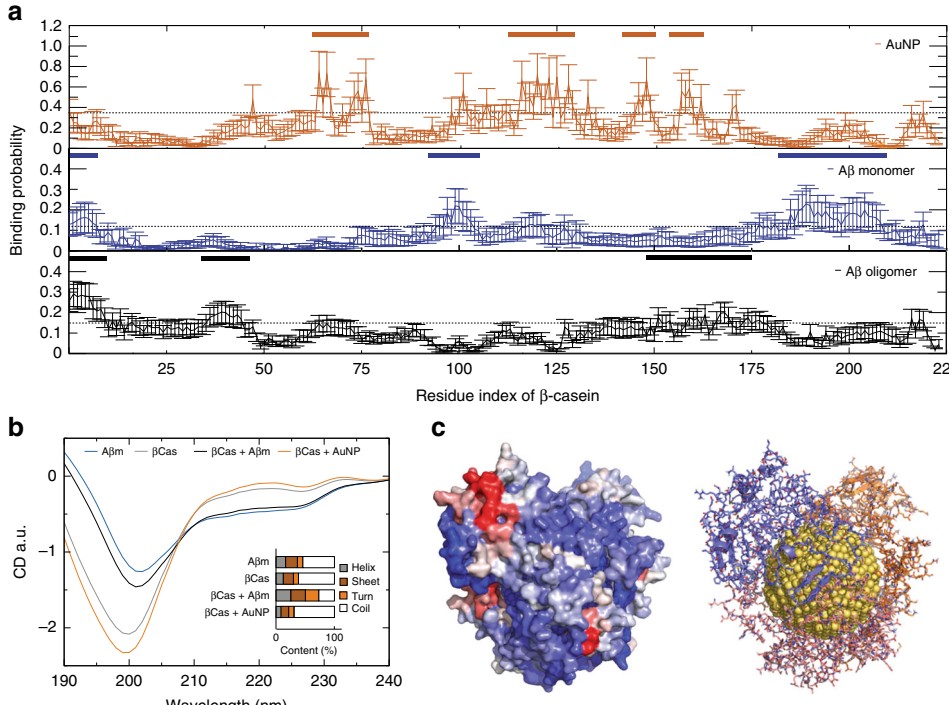

**Fig. 2** DMD simulations of βCas binding with AuNP, Aβ monomer, and oligomer. **a** Binding probabilities of βCas with an AuNP, an Aβ monomer and an Aβ oligomer, where high-binding is defined as residues with binding probabilities above one standard deviation from the average (dash lines). βCas high-binding residue regions with AuNP and with Aβ monomer/oligomer are highlighted with bars (inset). **b** Predicted CD spectra of secondary structure contents (inset) derived from simulations. **c** Predicted βCas-AuNP corona structures comprised of three βCas proteins on an AuNP surface (right) and corresponding molecular surfaces of the proteins (left) are shown to highlight their binding with an Aβ monomer, where each βCas residue was colored from purple (low) to red (high) according to its binding probability with Aβ monomer as in panel A middle. Source data are provided as a Source Data file

indicate that the βCas-AuNP complex could bind Aβ oligomer (Fig. 2a) and the strong binding between βCas and Aβ oligomer (Supplementary Fig. 5D) inhibited further growth of the initial oligomer (i.e., inset of Supplementary Fig. 5E) into an extended β-sheet structure via conformational rearrangements. Taken together, our simulations were not only consistent with the ensemble measurements in vitro, but also uncovered the molecular mechanism for the formation of the βCas AuNP complex and their inhibition of Aβ aggregation via either sequestering of Aβ monomers or capping of Aβ fibril elongation. In addition, the binding of Aβ with a bare AuNP is presented in Supplementary Fig. 6. Approximating the binding affinity and energy differences between the complex (βCas + AuNP, Aβ + AuNP) and individual components (βCas, Aβ) from DMD simulations (Fig. 2 and Supplementary Fig. 6) revealed that the binding of βCas with the AuNP was significantly stronger than the binding of Aβ with the AuNP ($\Delta\Delta G \sim -194$ kcal mol$^{-1}$) (Supplementary Table 2). Hence, replacement of βCas corona with Aβ was energetically unfavorable.

**Development of Aβ toxicity model in zebrafish larvae.** Zebrafish larvae express human orthologues of Aβ, APP, and γ-secretase components (PSENEN[37], NCTN[38], APH$_1$b[37]) 24 h after hatching[39]. Gene knockout or chemical inhibitors may create an imbalance among these protein components to result in neurological and behavioral abnormalities[40,41]. Here, an Aβ toxicity model was developed using zebrafish larvae (5 days old) by injecting Aβ into the cerebroventricular space (Fig. 3a and Supplementary Fig. 7). In vivo oligomerization of Aβ into toxic oligomeric species induced pathological features in zebrafish larvae after 5 days of Aβ treatment (Fig. 3b). Different concentrations of Aβ were injected into the larvae and no lethality was observed even with the highest concentration of Aβ, at 1200 fM per larva. However, reduced locomotion of the larvae was notable in a concentration dependent manner, with nonresponsive mobility and a loss of balance at higher Aβ concentrations (≥75 fM) (Fig. 3c). The nonresponsiveness of the larvae was recorded using tapping as a stimulus and loss of balance was observed as a tilt of the larvae from the normal horizontal axis to the imbalanced vertical axis (Supplementary Videos 1–3). The larvae injected with 10, 50, and 100 fM Aβ were characterized on an automated zebrafish behavior analysis system, to quantify total distance traveled and frequency of movement during the 1 h recording period. Observations were made on the third (Supplementary Fig. 8A) and fifth (Fig. 3d) day post treatment with Aβ. Significant reductions in both total distance traveled and frequency of movements were noted, in a concentration-dependent manner on the fifth day, in Aβ-treated larvae compared to untreated control. To visualize the presence of Aβ fibrils in the brain of the larvae, Congo red dye was injected in the cerebroventricular space of zebrafish larvae (Fig. 3e) on the third (Supplementary Fig. 8B) and fifth day post injection of Aβ (Fig. 3f). Significantly increased fluorescence was observed from the brain of the Aβ-injected larvae on the fifth day post injection. By comparison, Congo red dye was injected in untreated control and no fluorescence was observed (Fig. 3g). Aβ amyloid formation in the brain of zebrafish larvae was further confirmed by matrix assisted laser desorption/ionization (MALDI) analysis. Five days post injection with Aβ, the larvae heads were excised after euthanization, homogenized in phosphate-buffered saline (PBS) buffer and analyzed by MALDI. A peak corresponding to the molecular weight of Aβ was observed at 4538.1 mz$^{-1}$ (Fig. 3h). Aβ treated larvae were further fixed, cryo-sectioned and stained with Congo red. Aβ plaques were observed using the red fluorescent protein (RFP) channel

of a microscope (Fig. 3i). No red fluorescence was observed in untreated control (Fig. 3j).

**Biodistribution of βCas AuNPs in zebrafish larvae.** The biodistribution of βCas AuNPs was characterized by conjugating the AuNPs with NR dye and injecting the AuNPs via the intracardiac route (Fig. 4a). Whole-mount imaging was performed under the RFP channel of a fluorescence microscope at 0.5, 6, and 12 h after injection in order to trace the biodistribution of NR-βCas AuNPs in different regions of the larvae (Fig. 4b). No fluorescence was observed from the dorsal or lateral view of the larvae in the control (Supplementary Fig. 9). The zebrafish BBB is a double-layered membrane separating cerebral blood vessels from brain tissues. Alongside tight junctions, zebrafish BBB expresses occluding, claudins and p-glycoproteins and thus possesses a selectivity against xenobiotics[42,43]. In the present study, upon intracardiac injection of NR-βCas AuNPs into larvae, bright red fluorescence was observed from the brain after 0.5 h, indicating translocation of βCas AuNPs across the BBB (Fig. 4c). At 6 h after injection, the fluorescence from the cerebral region was decreased while it was recorded in the liver. However, the fluorescence was diminished from the liver at 12 h. βCas AuNPs were detectable by HSI and the SPR signals of the AuNPs were recorded from the brain sections of the larvae, prepared 0.5 h post injection of βCas AuNPs (Fig. 4d). However, no AuNPs or SPR were detected from the brain of untreated control. Finally, inductively coupled plasma mass spectroscopy (ICP-MS) was performed to further quantify the presence of AuNPs in the brain. The larvae treated with intracardiac injection of βCas AuNPs were euthanized at 0.5, 0.6, and 12 h and their heads were homogenized and quantified for Au in the brain and trunks. The concentration of Au was the highest in the brain at 0.5 h while decreased to the lowest level at 12 h (Fig. 4e). A correlation of AuNPs injected in the heart and delivered across the brain is shown in Fig. 4f. Injection of 1.5, 3, and 6 ng Au equivalent AuNPs via the heart delivered around 0.15, 0.45, and 0.5 ng of AuNPs to the brain, indicating that injection of >3 ng of AuNPs did not increase the delivery of AuNPs to the cerebral region. TEM images of microtome slices of the zebrafish larval brain also showed the presence of βCas AuNPs in the intracellular space (Supplementary Fig. 10).

Apart from the brain, 5 nm βCas AuNPs (conjugated with NR dye) also distributed to the fins and were imaged while circulating inside the microvasculature of zebrafish larvae (Supplementary Fig. 11). In contrast to βCas AuNPs, NR-conjugated βCas micelles (4.5 ng) were not able to translocate across the BBB and, instead, accumulated in the liver 6 and 12 h post injection (Supplementary Fig. 12) due to their larger sizes.

**Mitigation of Aβ toxicity and pathological symptoms.** Mitigation of Aβ toxicity was first assessed in vitro with SH-SY5Y neuronal cells. βCas AuNPs were able to sequester Aβ toxicity against SH-SY5Y cells in the viability assay (Supplementary Fig. 13A). Helium ion microscopy (HIM) revealed morphological damage induced by Aβ to the SH-SY5Y cells and their recovery by βCas AuNPs (Supplementary Fig. 13B, C). For in vivo, Aβ toxicity was induced in the zebrafish larvae by cerebroventricular injection of Aβ and was relieved by intracardiac injection of βCas AuNPs. Specifically, βCas AuNPs were administered at different time intervals post Aβ injection and the exposed larvae were studied for their behaviors 3 (Supplementary Fig. 14) and 5 days (Fig. 5a) post Aβ injection. βCas AuNPs completely relieved the symptoms when treated within 2 h of Aβ injection, as indicated by the total distance traveled, movement frequency and trajectories during 1 h of observation. Administration of βCas AuNPs, 6 h after Aβ treatment, partially alleviated the behavioral

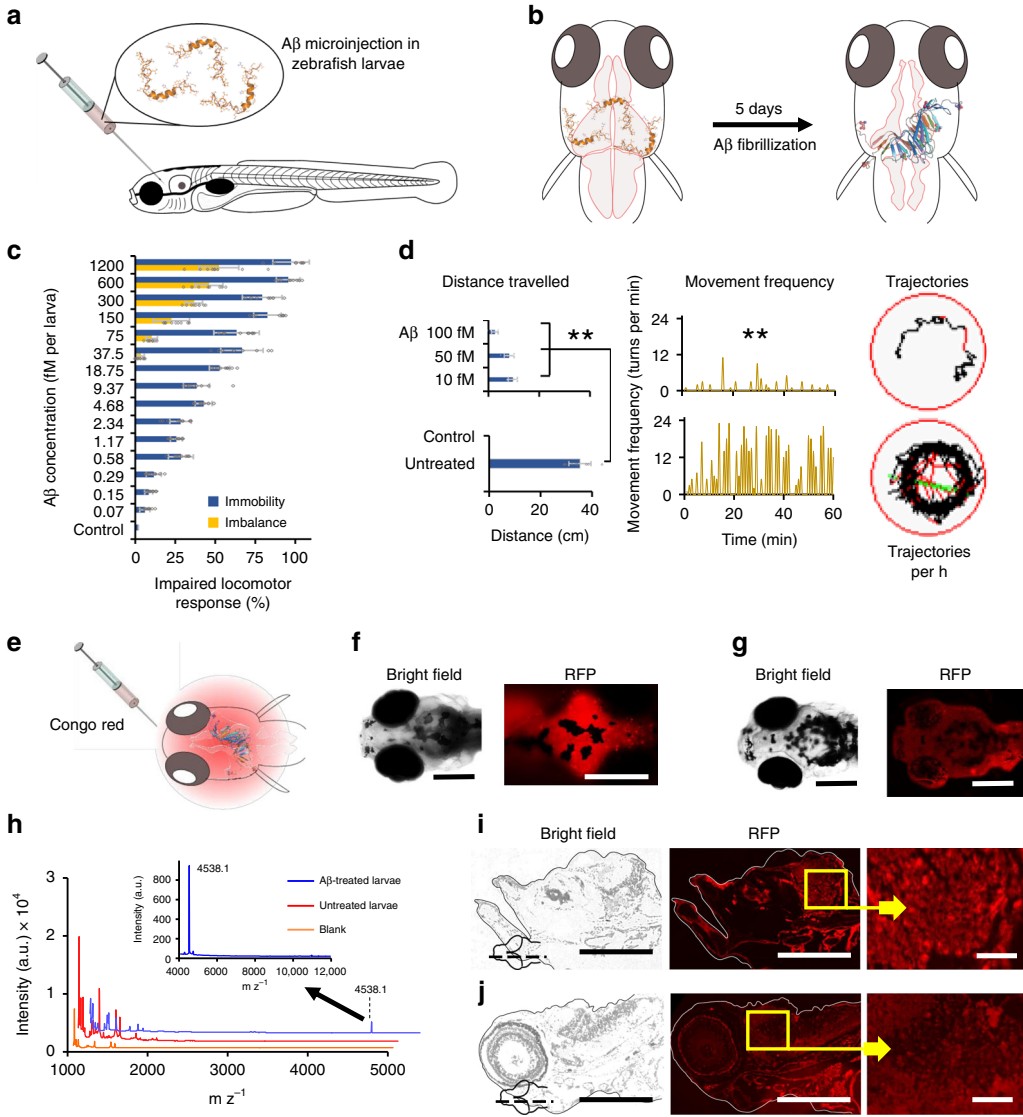

**Fig. 3** Aβ toxicity in zebrafish (*Danio rerio*) larvae. **a** Five-day-old zebrafish larvae were treated with Aβ and developed pathological symptoms (**b**). **c** Disruptive locomotive behavior was recorded in terms of percentage of larvae who failed to respond upon tapping (blue bars) and unable to maintain horizontal swimming position (orange bars) ($n = 10$). **d** The behavior of the larvae was further recorded on an automated zebrafish behavior monitoring system for 1 h at 5 days post Aβ treatment ($n = 10$). Total distance traveled along with movement frequency was significantly decreased compared to untreated control ($p < 0.005$). Representative trajectories of 100 fM Aβ treated and untreated larvae inside a single well of a 96-well plate, during 1 h of observation. **e** Five days after Aβ (100 fM) treatment, larvae were further treated with Congo red (100 fM) via cerebroventricular injection. Whole mount larvae were imaged under the RFP channel 6 h after Aβ treatment. **f** Significant fluorescence was retained in the cereberal region of larvae on the fifth day post Aβ treatment. **g** Congo red injected in untreated larva was not retained in the cerebral region. **h** MALDI detection of Aβ in the brain of zebrafish larvae, 5 days post Aβ injection ($n = 10$, Mean ± SD). Five days after cerebroventricular injection of Aβ, the heads of zebrafish larvae were excised after euthanization. The heads were homogenized in Holtfreter's buffer and subjected to MALDI-TOF/TOF analysis. Peak corresponding to Aβ molecular weight was observed at 4538.1 mz⁻¹. Untreated larvae and matrix alone were used as controls. **i** Congo red-stained thin section (sagittal) of Aβ treated larvae brain tissue. Bright red spots were observed in the cerebral region of larvae, corresponding to the Aβ amyloid or plaque formation. **j** In thin sections of the brain tissue of untreated larvae (negative control), no red spots were observed. Scale bars in all images are 200 μM, while in **i** and j inset are 20 μM. Error bars represent the standard deviation. Source data are provided as a Source Data file

symptoms. However, treatment with βCas AuNPs 12 h after Aβ injection did not rescue the larvae from Aβ toxicity, indicating the neurotoxicity of Aβ had been initiated. This observation correlates with the nucleation and oligomerization of Aβ into toxic species around 12 h, as indicated by the ThT kinetic assay (Fig. 1a). In contrast, βCas micelles failed to rescue the larvae from Aβ toxicity even injected 2 h after Aβ administration. βCas AuNPs and βCas as controls did not induce any behavioral abnormalities in zebrafish larvae (Supplementary Fig. 15). In

addition, citrate-capped AuNPs failed to rescue the larvae from Aβ toxicity, implicating that βCas, but not AuNPs was mainly responsible for toxicity mitigation (Fig. 5a).

Microtome slices of the brain tissues of zebrafish larvae, treated with Aβ and βCas AuNPs, were prepared on the fifth day post Aβ treatment and stained with Congo red. No amyloid plaque formation was observed (Fig. 5b), indicating elimination of Aβ species by βCas AuNPs. Immunohistochemistry (IHC) (Fig. 5c) and polarized light microscopy (Fig. 5d) further confirmed

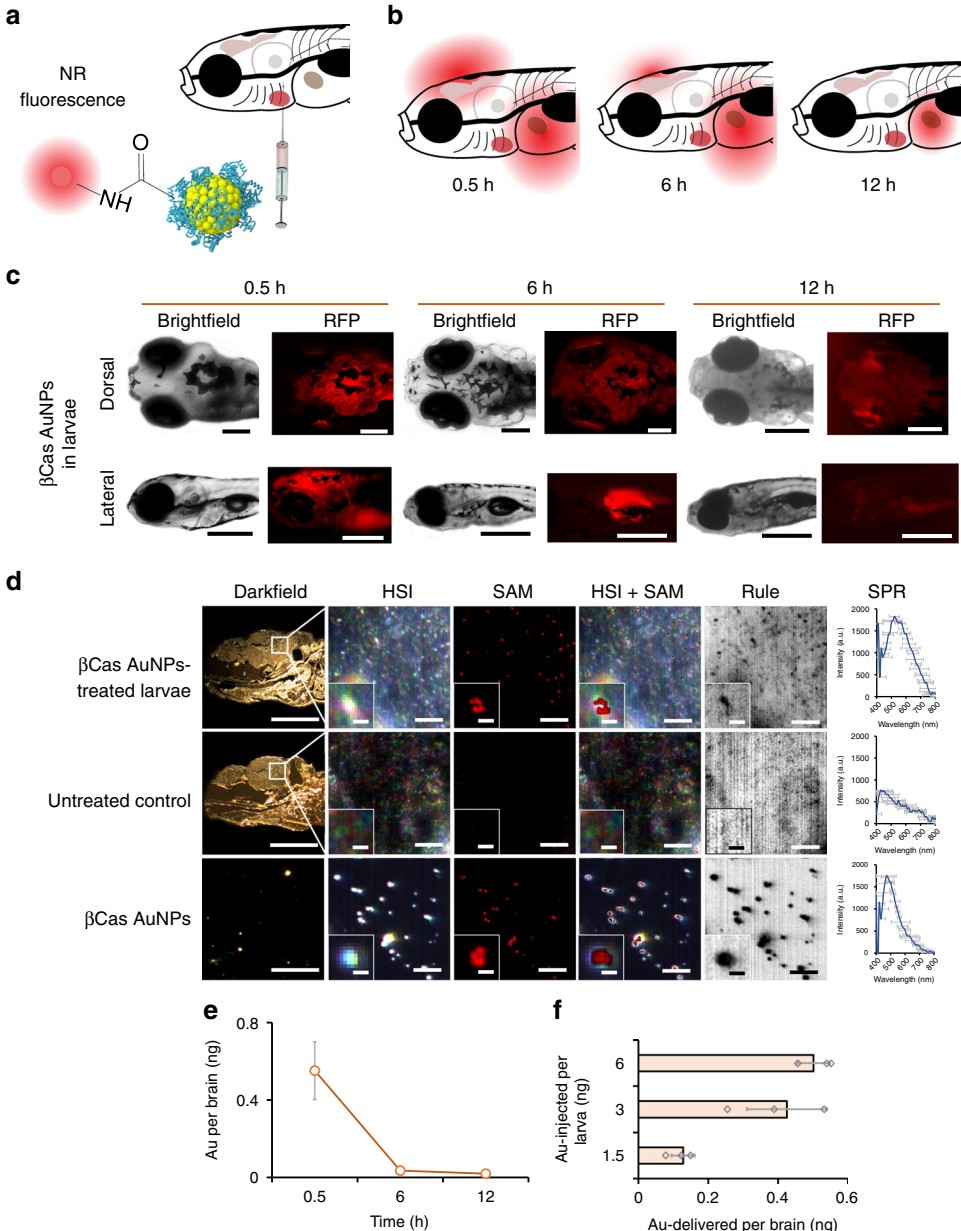

**Fig. 4** In vivo biodistribution of βCas AuNPs in zebrafish larvae. **a** βCas AuNPs were conjugated with neutral red and injected to zebrafish larvae (5 day old) via the intracardiac route in a dose equivalent to 3 ng Au per 4.5 ng βCas. **b** Labeled βCas AuNPs were traced for in vivo distribution at 0.5, 6, and 12 h in dorsal and lateral positions (**c**). Bright fluorescence was observed from the brain 0.5 h after AuNP administration. However, fluorescence was eliminated from the cerebral region in 6 h, while it took 12 h to eliminate from the body (scale bars: 200 μM). **d** Tissue microtome of zebrafish brain was subjected to HSI imaging. Spectral angular mapping (SAM) images were built from HSI by scanning against the βCas AuNPs spectral library. SAM and Rule images colored the pixels as red and black, respectively, that have matching spectra of βCas AuNPs. Zebrafish larvae with βCas AuNPs in the brain presented black spots in Rule images and red pixels in SAM images. SPR spectra with peak ~530 nm were observed in the brain of βCas AuNPs treated larvae. No such spectra were recorded for control larva (scale bars: darkfield 200 μM; HSI, SAM, HSI + SAM, Rule: 10 μM; inset scale bar: 2 μM; scale bar for βCas AuNPs: 10 μM, inset scale bar: 0.5 μM). **e** ICP MS analysis, where the AuNP concentration was the highest in the larval brain at 0.5 h, i.e., equivalent to 0.6 ± 0.1 ng of Au, and dropped to 0.05 ± 0.01 and 0.02 ± 0.008 ng at 6 and 12 h, respectively ($n = 10$). **f** Dose–response relationship between the amount of AuNPs injected vs. the amount of AuNPs delivered across the brain ($n = 10$). Significantly ($p < 0.05$) increased amount of Au was delivered when intracardiac dose of AuNPs was increased from 1.5 to 3 ng equivalent. However, increasing the dose from 3 to 6 ng did not improve AuNP delivery across the BBB, indicating a dose saturation. Error bars represent the standard deviation. Source data are provided as a Source Data file

deposition of Aβ amyloids in the brain tissues of zebrafish larvae, but not in βCas AuNPs-treated or untreated control larvae. The positive controls of fibrillized Aβ analyzed by IHC and polarized light microscopy were shown in Supplementary Fig. 16.

In addition to the behavioral symptoms, the neurotransmitters associated with Aβ toxicity and reactive oxygen species (ROS)

were quantified and loss of synaptophysin was imaged (Fig. 6), to vindicate the potency of βCas AuNPs against the toxicity of Aβ. Zebrafish are reported to possess cholinergic, glutamatergic and GABAergic neurotransmission that change in response to neurological dysfunction[44]. The acetylcholine esterase (AchE) and glutamate (GLT) levels were therefore assayed in Aβ and

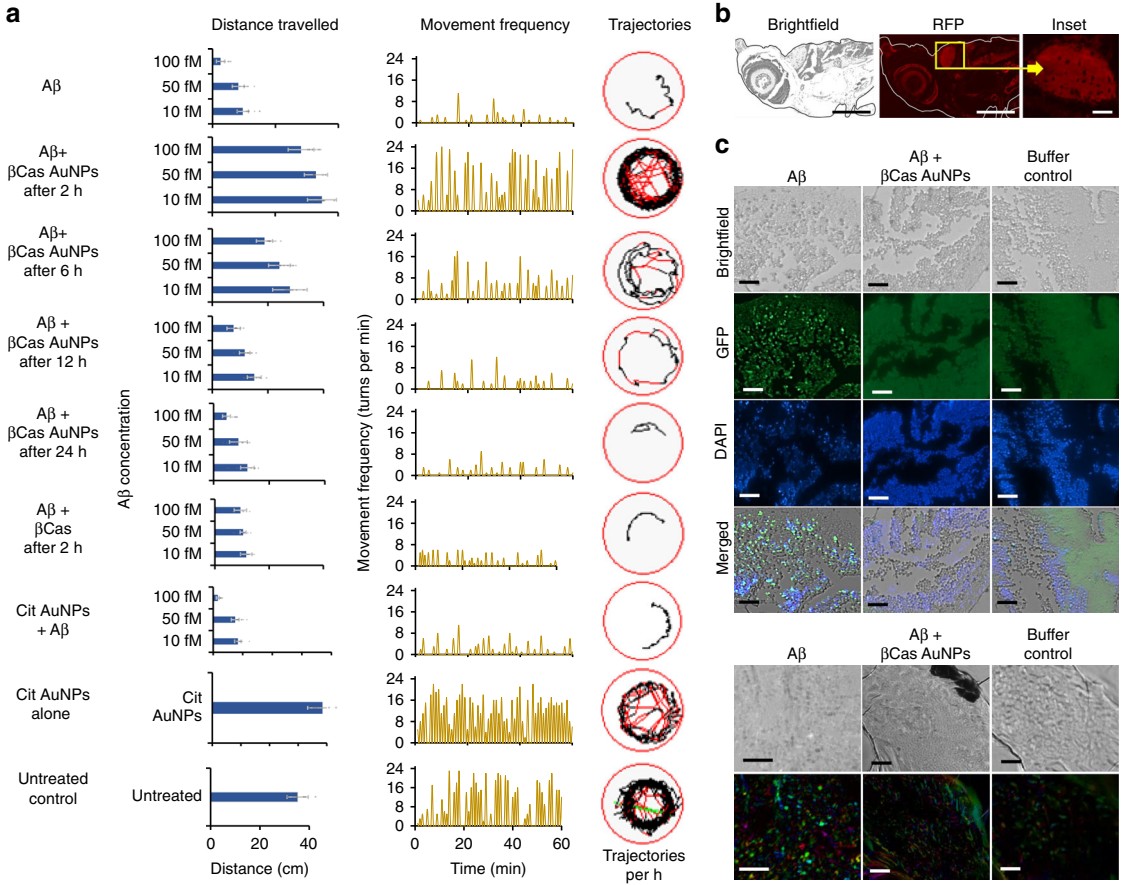

**Fig. 5** Mitigation of Aβ toxicity in zebrafish larvae with βCas AuNPs. **a** Aβ peptide was injected into the cerebroventricular space at 10, 50, and 100 fM concentrations ($n = 20$, mean ± SD). Zebrafish larvae were monitored on an automated behavior monitoring system at fifth day post Aβ treatment and parameters of total distance traveled, movement frequency and trajectory path were observed for 1 h. Significant ($p < 0.005$) difference in the behavior of the larvae was observed on the fifth day post treatment. βCas AuNPs injected, via the intracardiac route 2 and 6 h after the Aβ treatment, rescued the larvae from Aβ toxicity and from developing Alzheimer's-like symptoms. Representative trajectories of the larvae are displayed in the far-right column. Treating the larvae with βCas AuNPs, 12 and 24 h post Aβ treatment, failed to protect the larvae from developing Aβ toxicity. **b** Zebrafish larvae, treated with βCas AuNPs 2 h after Aβ treatment were fixed, sliced and stained with Congo red to image any Aβ fibrils that could have formed. Tissue slices of the brain section did not present any red fluorescence, indicating no Aβ fibril formation in βCas AuNPs treated larvae (scale bars: 200 μM; inset scale bar: 20 μM). Furthermore, immunohistochemistry (IHC) (**c**) and polarized light microscopy (apple green birefringence of amyloid) (**d**) revealed deposition of aggregated Aβ in the larval brain while no Aβ deposition was observed in βCas AuNPs or buffer treated larvae (Scale bars: IHC, 30 μM; polarized light microscopy, 50 μM). Error bars represent the standard deviation. Source data are provided as a Source Data file

Aβ + βCas AuNPs treated larvae. The heads were separated from the euthanized larvae, homogenized and used for the assays to minimize interference from the trunks. The biomarkers were first evaluated on the fifth day post Aβ (6 fM per larva) treatment. AChE levels in the Alzheimer's affected brain are known to be decreased[45], however, here no significant differences in the AChE activity ($0.10 ± 0.02$ a.u. per brain) or GLT level ($17.9 ± 1.2$ nm per brain) were observed compared to untreated control ($0.09 ± 0.01$ a.u. for AchE and $19.7 ± 1.2$ nm for GLT per brain) with 6 fM Aβ. As severe cases of Alzheimer's presence increased levels of AchE[46], the biomarker assay was performed with 600 fM Aβ and the AChE levels were found to increase twofold compared to the control, i.e., $0.22 ± 0.01$ a.u. AChE per brain (Fig. 6a). AChE levels were close to the control in the larvae treated with βCas AuNPs (3 ng Au equivalent) + Aβ (600 fM). βCas AuNPs and βCas, as controls, did not elicit any impact on the AChE levels. Similar results were observed for GLT, where βCas AuNPs reduced the GLT level from $29.2 ± 5.08$ to $18.1 ± 3.07$ nm per brain (Fig. 6b). According to the literature, the Aβ42 concentration in the gray and white matter of the brain of AD patients is 1.3 and 0.25 nm mg$^{-1}$, respectively[47]. Tg2576 mice models of AD present

~1600–1700 fM mg$^{-1}$ of Aβ42 after developing the disease symptoms[48]. However, the wet weight of whole zebrafish larva is ~1 mg and its brain is 4–5 times smaller than its body weight[49]. Considering this physiological relevance of the body weight, Aβ42 was injected in zebrafish larvae over a concentration range of 0.07–1200 fM per larva and based on locomotor response (Fig. 3c) 100 fM was selected for further experiments. However, 100 fM of Aβ concentration did not produce any difference in neurotransmitter levels that are usually disturbed in severe cases of AD[46]. Therefore, Aβ of 600 fM was used to observe any possible fluctuations in neurotransmitters.

ROS generation was quantified by a direct measurement of dichlorofluorescin diacetate (DCF) fluorescence from the larval brain (Fig. 6c, d). It has been shown in literature that oligomeric amyloid proteins directly interacted with cell membranes to induce cytotoxicity by membrane disruption and subsequent ROS generation[14]. Aβo were injected in the cerebroventricular space and their associated toxicity was determined by ROS generation, in comparison with the positive control of H$_2$O$_2$. The samples were mixed with DCF prior to microinjection in larvae. The corrected total (CT) fluorescence from H$_2$O$_2$ and Aβ treated

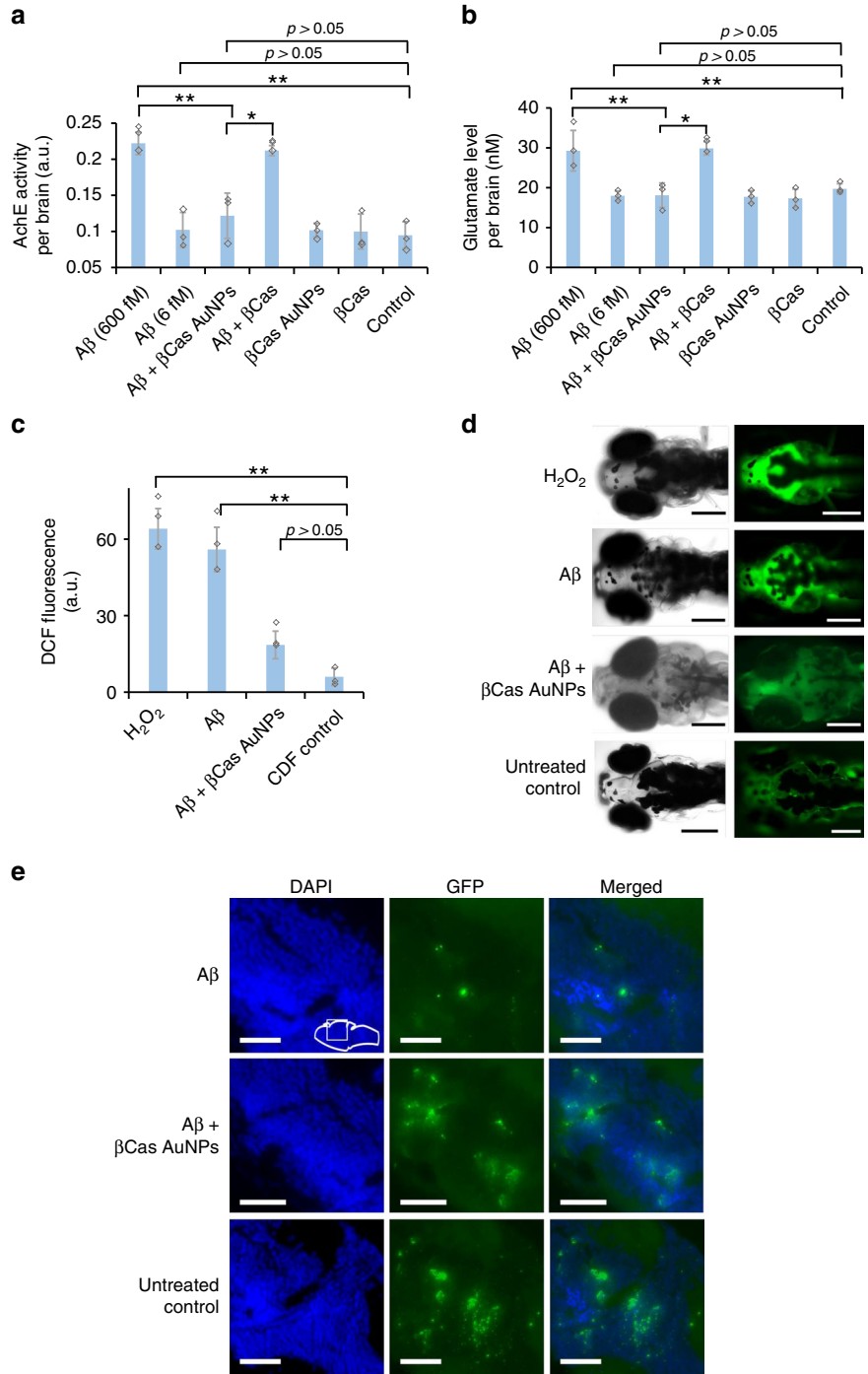

**Fig. 6** Neurotransmitters and reactive oxygen species (ROS) in the brain of Aβ treated larvae. The biomarkers were measured 5 days post Aβ treatment. Cerebrovascular injection of Aβ peptide at 6 fM per larvae did not significantly ($p > 0.05$) influence the AchE (**a**) and GLT (**b**) levels in zebrafish larvae ($n = 10$). However, increasing the Aβ dose to 600 fM significantly increased ($p < 0.05$) AchE and GLT levels. βCas AuNPs (3 ng Au per 4.5 ng βCas), injected 2 h post Aβ treatment, significantly ($p < 0.005$) reduced the AchE and GLT levels on the 5th day post Aβ (600 fM) treatment. βCas micelles (dose equivalent to βCas in 3 ng βCas AuNPs), in comparison, failed to improve ($p > 0.05$) the biomarker levels. **c** ROS generation was significantly ($p < 0.005$) high in Aβ treated (600 fM) larvae. ROS generation was supressed in βCas AuNPs treated larvae and close to control ($n = 10$). **d** Representative images of zebrafish larvae expressing DCF/ROS fluorescence when treated with $H_2O_2$, Aβ, Aβ + βCas AuNPs, and Aβ + βCas. **e** Larvae's brain sections were stained for synaptophysin. Aβ-treated larvae presented loss of synaptophysin indicating neurodegeneration. Scale bars in all images are 200 μM. Error bars represent the standard deviation. Source data are provided as a Source Data file

larvae were 41 ± 7.9 and 42.9 ± 8.7, respectively. However, upon treatment with βCas AuNPs, the CT fluorescence was reduced to 14.5 ± 5.4, comparable to DCF as negative control (8 ± 3.4). Synaptophysin-based neurodegeneration, an indicator for neuronal synapse, was also imaged via immunostaining (Fig. 6e). Aβ

treated larvae presented a significant loss of synaptophysin as compared to Aβ + βCas AuNPs or untreated control larvae.

In this study, zebrafish larvae is developed and used as a simple, in vivo visual model to study Aβ fibrillization, toxicity, behavioral pathology, neurodegeneration, and biodistribution

and nano-chaperone activity of βCas AuNPs. These advantages can be employed to screen or study the efficacy, pharmacokinetics and pharmacology of anti-Alzheimer's drugs, specifically nano-chaperone based therapeutic modalities. However, despite possessing a vertebrate nervous system, zebrafish larvae still develop cognitive and learning functions. Therefore, the behavioral pathology observed in this study may not be clinically equivalent to Alzheimer's symptoms. To study the Aβ toxicity and chaperone activity of βCas AuNPs, adult zebrafish was employed to offer a more advanced in vivo model with a cognitive capacity[50]. Microinjection of Aβ (1 μL, 50 μM) in adult zebrafish produced behavioral toxicity (Fig. 7a, b), Aβ aggregation in brain (Fig. 7c) and clinically relevant Alzheimer's-like symptoms (Fig. 7d–f). Retro-orbital microinjection (1 μL, 0.5 mM) of βCas AuNPs, 2 h post Aβ treatment, rescued the adult zebrafish from developing the cognitive dysfunction.

Expression of human orthologues of Aβ-associated neuronal machinery at 24 h post fertilization suggests suitability of zebrafish for AD modeling[39,51]. Macro-organization of the brain and cellular morphology of zebrafish are parallel to vertebrates and have led to studies of neurobehavioral pharmacology and stress-induced behavior[52]. In addition, exogenous microinjection or genetic overexpression of Tau in zebrafish has resulted in intracellular tangle formation and abnormalities in the animal's development and swimming behaviors[53,54]. These, together with our observations, support Aβ toxicity induction and AD modeling in zebrafish, especially within the context of cerebral deposition of Aβ and their associated behavioral pathology.

Intracardiac injection of βCas AuNPs mitigated the toxicity of cerebroventricularly injected Aβ₄₂ in a new, high-throughput zebrafish model. This remarkable capacity of eliminating toxic Aβo and rescuing the animal from AD-like symptoms was evidenced by in vitro assays of ThT, CD, and TEM, in silico

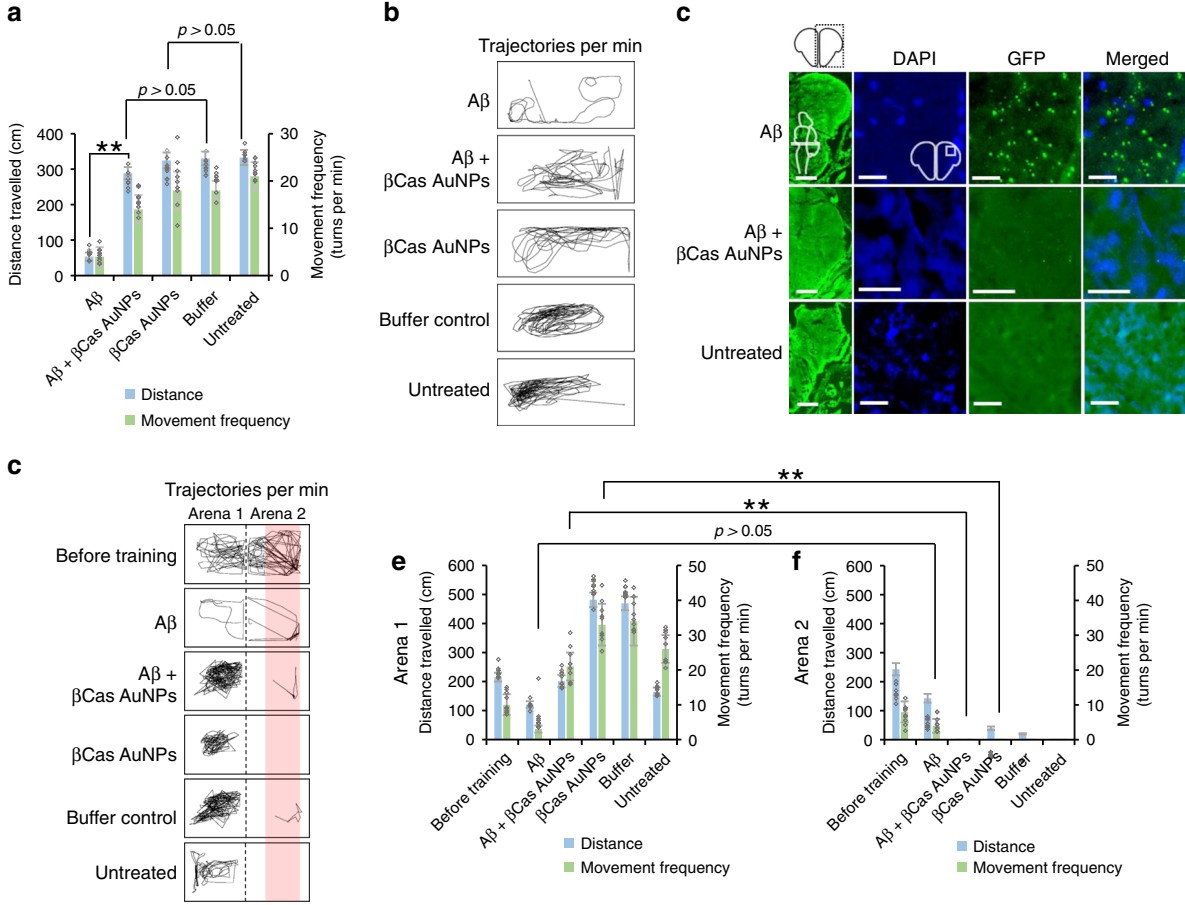

**Fig. 7** Mitigation of Aβ toxicity and Alzheimer's-like symptoms in adult zebrafish with βCas AuNPs. **a** Adult zebrafish (10 months old) were microinjected (cerebroventricular) with Aβ (1 μL, 50 μM) and observed for behavioral pathologies at 2 weeks post injection ($n = 4$, SD ± mean). To study the mitigation with βCas AuNPs, βCas AuNPs were microinjected (retro-orbital, 1 μL, 0.5 mM) 2 h prior to Aβ treatment. Aβ induced significant reduction in total distance traveled and movement frequency in adult zebrafish while βCas AuNPs were able to rescue the symptoms. Movement trajectories are presented in (**b**). Observations were made for 1 min, three times at a 2 h interval for each fish ($n = 3$). **c** IHC was performed on adult zebrafish brain sections to image the Aβ deposition. The first column represents the right cerebral brain of adult zebrafish in the GFP channel (Scale bars: 200 μM). DAPI, GFP, and merged images at higher magnifications revealed Aβ plaque deposition in Aβ treated but not in Aβ + βCas AuNPs, or untreated control (Scale bars: 20 μM). Furthermore, cognitive behavior of adult zebrafish was analyzed. **f** Zebrafish were trained to avoid swimming into the right half (Arena 2) of the swimming tank (1 L) that was labeled red and attached with a source of electric shock (9 V) (Supplementary Fig. 17). After training, the electric source was removed and the cognitive memory of the fish to remain in arena 1 and to avoid arena 2 was assessed for a period of 2 min ($n = 3$, three times for each fish at a 2 h interval). The movement trajectories of the fish in arena 1 vs. arena 2 are presented in panel F. Comparative analysis of distance traveled and movement frequency of the fish in arena 1 (**d**) vs. arena 2 (**e**) revealed cognitive dysfunction of Aβ-treated fish that were unable to avoid arena 2. However, βCas AuNPs treated, buffer control and untreated control fish were able to avoid swimming into arena 2. Error bars represent the standard deviation. Source data are provided as a Source Data file

examination of Aβ-βCas AuNP binding, ex vivo assays of microtome, HSI, ICP-MS, and MALDI analyses of Aβ impaired zebrafish brain, in vivo assays of ROS, behavior and neurological dysfunction biomarkers of zebrafish larvae, and cognition of adult zebrafish. The binding between Aβ and βCas was mediated by nonspecific interactions via the residues of the whey protein that were free from engagement with the AuNP surface. Adsorption of βCas onto an AuNP surface enabled trafficking of βCas across the zebrafish larvae BBB where βCas was then capable of efficiently binding Aβ for elimination. Furthermore, βCas AuNPs induced no harmful effects on the development of healthy zebrafish, owing to the biocompatibility of both the chaperone-like protein and the AuNPs. As this nano-formulation meets all key criteria for a potent in vivo amyloidosis inhibitor, it holds the promise to be further developed into safe-to-use, preventative nanomedicines against the pathologies of AD and other debilitating human amyloid diseases. Although zebrafish lack advanced cognitive capacity as possessed by rodents, they can serve as a robust, economic, and high-throughput alternative to complement neurological mouse models that are no longer deemed sufficient[19].

## Methods

**Animal husbandry and ethics statement**. The AB wild-type zebrafish (*Danio rerio*) was maintained in a fish breeding circulatory system (Haisheng, Shanghai, China) at $28 \pm 0.5$ °C with a 14 h light: 10 h dark cycle. The embryos were produced by adult spawning. For spawning, two pairs of male and female were placed in tank with shallow water. The male and female were separated by a removable partition and kept overnight. The spawning was triggered by removing the partition with first light in the morning and embryos were collected 2 h later, washed with 0.5 ppm methylene blue and placed in petri dish with Holtfreter's buffer. The healthy embryos at the same developmental stage were selected and developed to 5-day-old larvae for further measurement. All the experiments were performed with larvae in Holtfreter's buffer[55]. Tricaine (0.4% in Holtfreter's buffer) was used for anesthesia. When required, larvae were euthanized by placing in 0.01% tricaine in Holtfreter's buffer for 10 min that was pre-chilled at 4 °C. The excision of head from the trunk was performed under an optical stereomicroscope with a sharp surgical blade. All zebrafish experiments were performed in accordance to the ethical guidelines of Tongji University and the protocols were approved by the Animal Center of Tongji University (Protocol #TJLAC-019-113). All in vitro and other experiments were performed in compliance with the relevant ethics, laws, and institutional guidelines of Monash University Occupational Health & Safety.

**Synthesis of βCas AuNPs**. HAuCl$_4$ (1 mM, PBS pH 6) was heated at 40 °C and while stirring at 1000 rpm, added with an equal volume of βCas in PBS (pH 6) at the concentration of 1 mg mL$^{-1}$. The heating continued for 10 min and 100 μL of 0.5 mM NaBH$_4$ was added dropwise to the reaction mixture. The solution turned wine red indicating the formation of βCas AuNPs. Heating was stopped after 2 h while the reaction was kept on stirring for overnight. βCas AuNPs were purified by centrifugal filtration with 100 kDa spin filters. AuNPs were washed 3× with deionized water, transferred to PBS (pH 7.4) and stored at 4 °C for further experimentation. After purification, the βCas concentration in βCas AuNPs was determined by the BCA method[56]. Citrate-capped AuNPs (Cit AuNPs) were synthesized via a reported method[57]. Briefly, a solution of HAuCl$_4$ (10 mL, 1 mM) was brought to boiling at 120 °C and then 1 mL (10 mM) trisodium citrate was added dropwise. The solution turned to wine red, indicating the formation of Cit AuNPs. The solution was brought to room temperature, purified via centrifugal washing with deionized water and stored in dark for further use.

Neutral red (NR) dye was conjugated to βCas and βCas AuNPs for tracing their biodistributions in zebrafish larvae. The dye was conjugated via 1-ethyl-3-(3-dimethylaminopropyl)carbodiimide (EDC) coupling. Briefly, 2 mL of 6.25 μM of βCas or equivalent concentration of βCas AuNPs was stirred overnight with 18 μM of EDC, 23 μM of N-hydroxysuccinimide (NHS), and 14 μM of the NR dye. The NR-conjugated βCas or βCas AuNPs were purified via centrifugal filtration, transferred to PBS (pH 7.4) and stored for in vivo assays.

**ThT kinetic assay**. Aβ was treated with hexafluoro-2-propanol (HFIP) to break-down pre-existing small aggregates. Specifically, 0.5 mg of Aβ (Anaspec Inc., purity ≥ 95%) was dissolved in 500 μL of HFIP, aliquoted to different concentrations and dried to evaporate the HFIP. The dried Aβ was dissolved in 0.01% NH$_4$OH for dissolution purpose and left in the open for 20 min to evaporate NH$_4$OH, leaving behind the aqueous solution of Aβ that was used for further experiments. For the thioflavin T (ThT) assay, a 50 μL aqueous solution of 50 μM Aβ and 100 μM of ThT were incubated with or without βCas (6.25 μM) or equivalent concentration of βCas AuNPs in a 96-well plate. The ThT fluorescence

was recorded with excitation at 445 nm and emission at 488 nm, at 30 min intervals for 48 h.

**Transmission electron microscopy**. Aβ was mixed with βCas or βCas AuNPs in the same ratio as for the ThT assay and incubated for 48 h. After incubation, a drop of each sample was placed on a glow discharged, carbon coated copper grid and blotted after 1 min The sample-coated grid was negatively stained with 1% uranyl acetate that was blotted after 30 s. The grid was dried in vacuum and visualized with a Technei F20 transmission electron microscope operated at a voltage of 200 kV.

**CD spectroscopy, hydrodynamic size, and zeta potential**. The secondary structural contents of βCas, βCas AuNPs, Aβm, Aβo, Aβ fibrillized w/o βCas or βCas AuNPs were determined by CD spectroscopy. In all, 200 μL of the sample was pipetted into a CD cuvette and the concentration of Aβ was 100 μM while βCas and βCas AuNPs were 12.5 μM βCas equivalent in all CD measurements. The incubation time was 48 h at 37 °C and the incubation medium was deionized water. CD spectra were recorded from 190 to 240 nm with a 1 nm step size. The acquired data were presented in the unit of millidegrees (mdeg). Percentage secondary structure contents were determined by analyzing the CD data via Dichroweb with Contin/reference set 4[58]. The hydrodynamic diameter and zeta potential of the AuNPs were measured with dynamic light scattering under ambient conditions (Malvern Instruments). The concentrations and incubation conditions were the same as for the CD experiments.

**Binding capacity (BCA, TGA, TEM, CD, UV-SPR, and fluorescence microscopy)**. The binding capacity between Aβ and βCas or βCas AuNPs was determined via a BCA assay[56] and TGA. Briefly, 6.25 μM of βCas or equivalent βCas AuNPs were incubated with varying concentrations of Aβm (monomers) or Aβo (oligomers) ranging from 2 to 1137 μM. Incubation time was 48 h at 37 °C in deionized water. HFIP-treated Aβ was considered as Aβm while Aβm incubated in water for 12 h at 4 °C was considered as Aβo. After incubation, the samples were centrifuged at $17,300 \times g$ for 30 min and the pellets were redispersed in 10 μL of deionized water and subjected to BCA protein content quantification. The BCA binding efficacy was presented in terms of the amount of Aβ bound to βCas or βCas AuNPs. For TGA analysis, the pellet dispersed in 10 μL water was placed as a drop on a platinum pan. The samples were held at 80 °C for 30 min and then scanned from 80 to 800 °C at a scanning rate of 10 °C min$^{-1}$ under a constant flow of nitrogen of 1 mg mL$^{-1}$. For Aβm and Aβo, 10 μL of 1 mg mL$^{-1}$ of the peptide was placed on a TGA pan and scanned under the same conditions.

To assess binding affinity, βCas AuNPs or NR-βCas AuNPs (12.5 μM βCas equivalent) were incubated with preformed Aβo or Aβm (100 μM) for 3 h at 37 °C and unbound Aβm/o was removed via centrifugal washing thrice ($25,000 \times g$ for 10 min at 4 °C). CD and TEM were performed as described above in the respective sections. UV-SPR for βCas AuNPs and fluorescence spectra (NR-βCas AuNPs, excitation at 470 nm) were recorded with a microplate reader.

**Cellular toxicity**. SH-SY5Y (ATCC® CRL-2266™) human bone marrow neuro-blastoma cells were cultured in Dulbecco's Modified Eagle Medium: Nutrient Mixture F-12 (DMEM/F12) with 10% fetal bovine serum (FBS). A 96-well plate (Costar black/clear bottom) was coated with 70 μL of poly-L-lysine (Sigma, 0.01%) and incubated at 37 °C for 30 min. After removing poly-L-lysine, the wells were washed by PBS thrice. Cells (~50,000 cells per well per 200 μL medium) were added to the wells and incubated at 37 °C with 5% CO$_2$ for 24 h to reach ~70–80% of confluency. The cell culture medium was then refreshed with 1 μM propidium (PI) dye in DMEM/F12 with 10% FBS and incubated for another 30 min Aβ was freshly dissolved in 0.005% NH$_4$OH buffer, in the presence or absence of βCas AuNPs and added to the wells with final concentration of 20 and 50 μM for Aβ and βCas AuNPs, respectively. Cellular toxicity was recorded by Operetta (PerkinElmer, 20× PlanApo microscope objective, numerical aperture: 0.7) in a live cell chamber (37 °C, 5% CO$_2$) after 15 h of treatment. The percentage of dead cells (PI-positive) to total cell count was determined by a built-in bright-field mapping function of Harmony High-Content Imaging and Analysis software (PerkinElmer). The measurement was performed in triplicate and conducted at five reads per well. Untreated cells were recorded as control.

**Helium ion microscopy**. SH-SY5Y neuronal cells were incubated with Aβ in the presence or absence of βCas AuNPs as described for the cellular toxicity assay. The incubation was performed for 2 h at 37 °C and then stabilized by 2.5% paraformaldehyde. The samples were incubated at 4 °C overnight. The paraformaldehyde/medium was replaced with gradient concentrations of ethanol in the five steps of 20%, 40%, 60%, 80%, and 95%, respectively, with ~2 h of rest time at each gradient. In all, 30 μL suspension of cells was air-dried on a carbon tape and the morphologies of the cells were visualized by HIM (Orion NanoFab, Zeiss, USA). Untreated cells were used as control.

**Microinjection of Aβ, βCas, βCas AuNPs, and Cit AuNPs in zebrafish larvae**. HFIP-treated Aβ (10 μg) was dissolved in PBS (pH 7.4) to make a stock solution of

100 μM. Dilutions of 0.07–1200 fM of Aβ per 5 nL were made in PBS and injected (5 nL injection volume) into the cerebroventricular space of 5 days old zebrafish larvae. PBS alone was used as negative control. For microinjection, zebrafish larvae were anesthetized by adding 2 drops of 0.4% tricaine in petri dish and waited until the larvae stopped moving in response to tapping on the table. The larvae were positioned on a 1% agarose gel plate and microinjected with Aβ peptide. Micro-injections were performed with a fine calibrated needle of a pneumatic micro-injection system (PV830 Pneumatic Picopump, WPI) operated under 20 psi of injection pressure. The tip of the glass capillary needle was inserted in the ventricular space, across the dorsal soft skin tissue. The tip was ensured not to penetrate more than 0.1–0.3 mm across the center meeting point of left and right telencephalon (Supplementary Video 4). βCas and βCas AuNPs were administered under similar conditions via intracardiac microinjection (Supplementary Video 5). The original as-synthesized βCas AuNPs solution contained 1 and 1.5 ng of Au and βCas per 5 nL. The original βCas AuNPs solution was concentrated 3× and redispersed in PBS. A 5 nL of this solution was microinjected into zebrafish larvae via the intracardiac route and each 5 nL contained 3 ng of Au per 4.5 ng ($37.5\,\mu M\,mL^{-1}$) of βCas in the form of βCas AuNPs. βCas solution of equivalent concentration was prepared in PBS for microinjection. For dose dependent delivery of βCas AuNPs in cerebral tissues, the original βCas AuNPs solution was concentrated to 1.5, 3, and 6×, and dispersed in PBS prior to intracardiac microinjection to zebrafish larvae. Cit AuNPs were concentrated to equal concentration as βCas AuNPs and microinjected into the larvae with the same protocol.

**Zebrafish larvae behavioral pathology.** Larval response to tapping stimuli in a 96-well plate was observed. The 96-well plate was tapped gently at the rate of 1 per sec and the larvae unable to move after five consecutive stimuli were counted as nonresponsive. Furthermore, the larvae losing their horizontal swimming position at higher doses of Aβ were also counted and the percentage of the larvae losing response to stimuli and their swimming position was calculated. The swimming behavior of zebrafish larvae was observed with an automated zebrafish behavior recording system ZebraBox (Viewpoint) and characterized in terms of total distance traveled by the larvae in a 96-well plate and range of the movement that was >90°, clockwise or counter clockwise, were counted. Representative trajectories of the movement were also recorded by built-in sensors. The observation period was 1 h. The number of larvae in each group was 20 and 3 groups were used for each sample. The larvae treated with βCas or βCas AuNPs, 2 h after Aβ treatment, were monitored for behavioral pathology with the same method.

**Fluorescence imaging, IHC, and polarized light microscopy.** Whole mount larval imaging was performed under the brightfield (BF) and RFP channels of a stereomicroscope (Olympus MVX10). For imaging of Aβ in the cerebral region, the Aβ treated larvae (100 fM, cerebroventricular microinjection) were microinjected with 100 fM of Congo red on the third and fifth day post Aβ treatment. The larvae were placed in Holtfreter's buffer for 6 h to allow staining of the amyloids. The larvae were anesthetized immediately prior to imaging with 0.4% tricaine, and positioned in dorsal or lateral view in a drop of 1% low-melting agarose gel. The cerebral region, fin and mid-vascular region of the larvae were imaged with the same method. The biodistribution of NR-conjugated βCas or βCas AuNPs was imaged by following the same method 0.5, 6, and 12 h post-βCas or βCas AuNPs treatment via intracardiac microinjection. The microinjection volume was 5 nL with a dose of 4.5 ng for βCas (37.5 μM) or βCas AuNPs equivalent to 4.5 ng βCas.

For Congo red staining of the sliced sections of the zebrafish larvae, Aβ (100 fM) treated larvae were first fixed in 2.5% paraformaldehyde for 12 h at 4 °C. Microtome slices of zebrafish larvae were prepared by embedding the larvae into paraffin. The paraffin-embedded larvae were cut into thin slices of 5 μM via microtome and slices were placed in a hot water bath (40 °C) for removal of any wrinkles. The slices were then mounted on glass slides and dried. Slide mounted slices were dewaxed by treating with (1) xylene for 2 h, (2) absolute ethanol for 15 min, and (3) 75% ethanol for 5 min and then rinsed with deionized water. The slices were treated with 0.5% Congo red stain in 50% ethanol for 20 min, rinsed with water, differentiated with 1% NaOH solution in 50% ethanol (5–10 dips) and again rinsed with water. The slices were then dehydrated with 95% ethanol (3 min) and 2 dips in 100% ethanol (each 3 min) and cleared with 2 dips in xylene (each 3 min). The samples were finally sealed with neutral gum and imaged with an optical microscope.

For IHC, 5 μM thick sections of zebrafish larvae were mounted on glass slides as for Congo red staining. The dried sections were washed in PBS (pH 7.4) and TritonX-100 (0.05% in PBS) for 5 min each. A drop (50 μL, 2 μg mL⁻¹) of primary antibody (Anti-amyloid β42, mouse monoclonal, Anaspec, AS-55922) was placed on each section on the glass slides and incubated at 4 °C overnight. Primary antibodies were washed away from the sections by dipping in PBS and TritonX-100 (0.05% in PBS) for 5 min. Sections were incubated with a drop (50 μL, 2 μg mL⁻¹) of secondary antibody (Goat anti-mouse HiLyte™ Fluor 488—labeled, Anaspec, AS-61057-05-H488) for 6 h at room temperature. The secondary antibodies were washed away by dipping the slides in TritonX-100 (0.05% in PBS) for 5 min. Slides were dried, mounted with a cover slip using a drop of 50% glycerol and imaged under a fluorescence microscope (Nikon Ti-Eclipse). The larvae's brain sections were immunostained for synaptophysin with the same method using primary (50 μL, 2 μg mL⁻¹ of anti-Synaptophysin antibody, abcam, Cat# ab32594) and

secondary antibodies (50 μL, 2 μg mL⁻¹ of Goat Anti-Rabbit IgG H&L, Alexa Fluor® 488, abcam, Cat# ab150081).

Polarized light microscopy was performed on the Congo red-stained larvae tissue sections. The slides were imaged for birefringence under an Abrio polarization microscope. A drop (50 μL, 20 μM) of fibrillized Aβ was placed and dried on a glass slide. The dried sample was processed same as for larva tissue sections for immunostaining and polarized light microscopy and used as control.

**Darkfield HSI.** Zebrafish larvae at 0.5 h post treatment with intracardiac microinjection of βCas AuNP (3 ng Au per 4.5 ng βCas equivalent) were euthanized and fixed in 2.5% paraformaldehyde for 12 h and then sliced to thin sections, dehydrated and mounted on glass slides with the same procedure as described for fluorescence microscopy. HSI was performed with a CytoViva darkfield microscope equipped with a pixelFly CCD camera. ENVI 4.8 software was used to capture and process the images and to acquire AuNPs spectra. The darkfield images were captured and then scanned for HSI for βCas AuNPs treated larvae, untreated control larvae and βCas AuNPs alone. βCas AuNPs alone as control were used to acquire the spectral library of AuNPs by selecting ~1500 pixels with region of interest function of ENVI 4.8. Spectral libraries obtained from βCas AuNPs and a mean spectral signature was generated. The mean spectral signature of βCas AuNPs was used to filter against the selected pixels (~1000) from βCas AuNPs treated and untreated control larval images to generate spectral angular mapping images. Rule images were obtained by matching the pixel spectra from βCas AuNPs against βCas AuNPs or untreated larval images. Rule images darkened the pixels with matching spectra of βCas AuNPs. All images were normalized against lamp spectra. For HSI imaging of βCas AuNPs and βCas AuNPs incubated with Aβ, the AuNPs were incubated with Aβ at the same ratio as for ThT for 48 h. A drop of each sample was placed on a glass slide covered with a slip. Darkfield images and SPR spectra of the AuNPs were obtained by scanning ~1500 pixels for each sample.

**TEM imaging of brain tissues.** For TEM analysis of tissues, zebrafish larvae were injected with 3 ng Au per 4.5 ng βCas equivalent βCas AuNPs via intracardiac microinjection as described above, euthanized and fixed in 2.5% glutaraldehyde 0.5 h post treatment. The larvae were treated with 1% osmium tetroxide for 4 h (4 °C) and then washed three times with 0.1 M PBS buffer (pH 7.4), 15 min each. The larvae were then treated with 1% citrate in 0.1 M PBS buffer (pH 7.4, 20 °C) for 2 h and again washed three times with 0.1 M PBS (pH 7.4), at 15 min each. After that the larvae were dehydrated by treating with 50, 60, 70, 80, 90, and 100% ethanol, at 15 min each. Following that the larvae were treated with acetone: 812 embedding agent (1:1) and then with 812 embedding agent, both for overnight and then baked at 60 °C for 4 h for polymerization of embedding resin. The larvae were finally sliced into ultra-thing sections of 60–80 nm with Diatome ultra 45°. The sections were double stained with 2% solutions of uranyl acetate and lead citrate, 15 min each, dried overnight and then imaged with a transmission electron microscope (Technei G2 20 TWIN) operated at 80 kV.

**Inductively coupled plasma-mass spectrometry.** Delivery of βCas AuNPs across the larvae's brain was quantified with ICP-MS analysis. Zebrafish larvae were microinjected with 1.3, 3, and 6 ng Au equivalent βCas AuNPs via the intracardiac route with the same method as described above. The larvae were euthanized 0.5, 6, and 12 h post microinjection and their heads were excised and homogenized in PBS (pH 7.4) in a Teflon-glass homogenizer (70 Hz for 1 min). Brain or trunk homogenate was made up to 1 mL with PBS (pH 7.4), added with 9 mL of 68% HNO₃ and digested by stepwise heating at 100, 150, 170, and 190 °C for 30, 30, 30, and 90 min, respectively. The dried and digested layer was dissolved in 1 mL of 4% HNO₃ and analyzed with ICP-MS (Agilent 7700) for quantification of Au. The instrument was operated under 0.75 MPa Ar pressure and a standard calibration was made with Au spiked PBS samples, digested in the same way as the larval samples. Untreated larvae and PBS alone were used as negative controls. The number of larvae in each group was 20 and 3 groups per sample were used for analysis.

**Matrix-assisted laser desorption ionization-time of flight mass spectrometry (MALDI TOF MS).** Saturated solution of sinapinic acid (SA) was prepared in ethanol and 1 μL of the sample was dried on a ground steel MALDI plate. Another saturated solution of SA was prepared in acetonitrile and trifluoroacetic acid (30:70, v/v) and mixed with the zebrafish larvae head homogenate at a 1:1 ratio. In all, 0.5 μL of this mixture was applied to previously dried SA layer. The dried layer was analyzed by a Bruker ultraflextreme MALDI-TOF/TOF in the linear positive mode. The instrument was calibrated with protein calibration standards I and II. A total of 8000 shots were gathered across the sample spots using Flexcontrol software (3.4) in the range of 1–20 kDa. The acquired spectra were processed by baseline subtraction and peak picking using Flexanalysis software (3.4).

**Biomarkers and ROS assay.** Acetylcholine esterase (AchE) and glutamate (GLT) levels were measured as biomarkers for neurodegeneration. Briefly, zebrafish larvae treated with Aβ, Aβ + βCas, and Aβ + βCas AuNPs were euthanized and their heads were excised from the trunks. The heads were homogenized in PBS (pH 7.4)

and AchE and GLT levels were estimated using assay kits according to reported literature[59,60].

For ROS assay, Aβ (100 fM) were injected to the cerebroventricular space of zebrafish larvae and 2′,7′-DCF (5 nL of 2 μM) was injected 5 days post Aβ treatment. The whole mount larvae were imaged under the green fluorescence protein channel of an optical microscope 1 h after DCF treatment. For βCas AuNPs, the nanoparticles were injected 2 h post Aβ treatment followed by the same procedure. H₂O₂ (5 nL of 0.1%) was used as positive control and injected 3 h prior to DCF microinjection in positive control larvae. DCF fluorescence was quantified by excising the head from the trunk of euthanized larvae, homogenizing in PBS buffer (50 μL, pH 7.4) and reading the DCF fluorescence with a microplate reader with excitation/emission at 495 nm/529 nm.

**Microinjection in adult zebrafish.** Adult zebrafish at the age of 10 mth were used for the cognition experiment. The fish was maintained, before and during the experiment, as described in the "Animal husbandry and ethics statement" section. For microinjection, adult fish were anesthetized with ice chilled tricaine (0.01% in Holtfreter's buffer for 20 s). Cerebroventricular microinjection of Aβ (1 μL, 50 μM) was performed via 1 μL Hamilton glass syringes. Aβ peptide was injected in between the right and left telencephalon and the needles did not penetrate more than 1 mm (Supplementary Video 6). The fish were held in place via a forcep. The syringes were washed with 70% ethanol and 1× PBS twice, in between the injections. For the group with Aβ with βCas AuNPs, βCas AuNPs were injected 2 h prior to the injection of Aβ. βCas AuNPs (1 μL, 0.5 mM, 20 psi injection pressure) were slowly introduced into the systemic circulation of the fish via retro-orbital microinjection (7 o'clock position), using a sharp glass capillary needle (Supplementary Video 7). The fish was placed back into the tank for recovery. The adult zebrafish were grouped (n = 3) and injected with Aβ, Aβ with βCas AuNPs, βCas AuNPs alone and PBS. Buffer injected and untreated fish were considered as controls.

**Behavioral pathology and cognitive function test of adult zebrafish.** The adult zebrafish microinjected with the above described samples were monitored on a daily basis for any apparent change to their swimming activity. The swimming activity of the fish started to change 1-week post treatment and became significantly apparent at 2 weeks post treatment. The behavioral pathology of the fish was recorded with ZebraBox (Viewpoint), using a 1 L fish tank, and characterized for total distance traveled and movement frequency. The recording was performed for 1 min, 3 times for each fish at a 2-h interval.

Cognitive function test was performed by hypothetically dividing the 1 L fish tank into two halves, i.e., arenas 1 and 2 (Supplementary Fig. 17). A red colored paper was attached to the bottom of the tank to associate a color with arena 2. Fish were allowed to freely swim in the whole tank for 30 min and then trained for 20 min to avoid swimming into arena 2 by using an electric shock punishment. Whenever the fish swam into the red arena 2, it was punished by dipping the electrodes of 9 V in electric potential. After 20 min of training, the electric source was removed and the cognitive ability of the fish to avoid arena 2, while swimming in arena 1 was recorded for 2 min. The comparative distance traveled and movement frequency of the fish in arena 1 vs. 2 were recorded simultaneously. The recordings were made 3 times for each fish (n = 3) at 2 h intervals. The results were analyzed via EthnoVision X1. The analysis parameters were as follows; animal: adult zebrafish, arena: open field square template (divided into two-halves for the cognitive function test), tracking feature: central point, sample rate: 5 per second, detection setting level: sensitive enough to track the fish in the whole tank, threshold for movement frequency: 50° turn (clockwise or counter clockwise) and minimum 0.5 cm of travel.

IHC was performed on adult zebrafish brain sections. Adult zebrafish at 2 weeks post Aβ or Aβ + βCas AuNPs treatment were euthanized by placing the animals in ice chilled tricaine (1% in Holtfreter's buffer for 1 min). The heads were separated from the bodies, at pectoral fin, with a sharp scalpel and fixed in 2.5% paraformaldehyde overnight. The heads were treated with 20% sucrose overnight, fixed in Tissue Trek OCT mounting medium at −20 °C and sectioned into 20 μM thick sections via cryostat. The sections were mounted on gelatinized glass slides, immunostained for Aβ as described above in the "Fluorescence imaging, IHC, and polarized light microscopy" section and imaged via a fluorescence microscope.

**Statistical analysis.** All the experiments in the manuscript were repeated three times, unless specified, and data was presented as mean ± SD. For the zebrafish experiments, 20 larvae per group and 3 group per sample were used to minimize experimental error. The significance of results was determined by one-way ANOVA followed by Turkey's test and p values less than 0.05 were considered significant (presented with * in figures) while p values less than 0.005 were considered as highly significant (presented with ** in figures).

**Reporting summary.** Further information on research design is available in the Nature Research Reporting Summary linked to this article.

## Data availability

Data supporting the findings of this paper are available from the corresponding authors upon reasonable request. Reporting summary of this article is available as Supplementary Information file. The raw data underlying the respective main text (Figs. 1–7) and Supplementary figures (Supplementary Figs. 1, 2, 8, 13, 14 and 15) are provided as Source Data File.

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

## Acknowledgements

This work was supported by ARC Project No. CE140100036 (Davis), the NSFC grant #21607115 and #21777116 (Lin), NSF CAREER CBET-1553945 (Ding), and NIH MIRA R35GM119691 (Ding). Javed acknowledges the support of Monash International Post-graduate Research Scholarship (MIPRS) and Australian Government Research Training Program Scholarship. TEM imaging was performed at Bio21 Advanced Microscopy Facility, University of Melbourne and polarized light microscopy was performed at Monash Micro Imaging, Monash University. Javed and Lin thank Shanghai Science and Technology Commission "Belt and Road" initiative program, Grant no. 17230743000. The authors acknowledge the support from Profs. Ting Xu and Daqiang Yin on the Zebrabox instrument.

## Author contributions

P.C.K., I.J., S.L. and T.P.D. designed the project. I.J. and P.C.K. wrote the paper. I.J. performed AuNP synthesis, TEM, CD, UV-SPR, fluorescence characterizations, larval, and adult zebrafish behavioral and cognitive assays, histology, and polarized light microscopy. I.J., M.Z. and T.Y. performed in vivo microinjection. I.J. and G.P. performed ICP-MS analysis. Y.X. and F.D. conducted DMD simulations. A.F. performed cell viability assay and HIM imaging. C.L.P. and A.K. provided inputs to the discussion of the paper. All authors agreed on the presentation of the paper.
