## [Peer Review File · Nature Communications]

Reviewers' comments:

Reviewer #1 (Remarks to the Author):

The submitted manuscript by Javed and colleagues outlines a novel method of using gold nano-particles coated with beta-casein to sequester amyloid beta-42 peptides both in-vitro and in the brain of zebrafish larvae. The main conclusion of the manuscript is that the nano-particles may have broad use for eradication of toxic amyloid proteins in a range of human diseases. They provide behavioral data, ROS analysis and neuronal dysfunction biomarker assays to support their main conclusion. My expertise is in the zebrafish model and Alzheimer's disease therefore most of my comments will be focused on these aspects of the manuscript. Please note I have no particular expertise in molecular dynamics algorithms or simulations.

Comments and queries:

- The Introduction mentions that both beta-casein and AuNPs both have chaperone like activity. Can the authors elaborate more on this activity and perhaps include more information in the introduction about this complex? Does this complex cross the BBB in humans?
- Further discussion is required on other amyloid diseases. There is still a huge debate on the role of "toxic" amyloid proteins in the AD, this should also be mentioned.
- I do not agree that amyloid beta is inducing a "Alzheimer's-like pathogenesis". There is no discussion in the manuscript on how the behavioral pathology is similar to clinical symptoms observed in this disease. I think the authors should consider changing this term and also whether this actually is an AD-like zebrafish model.
- The authors need to discuss the limitations of the model. This is a larval model and the fish are still developing so how does this relate to an aging disease in particular when looking at neurotransmitters in the zebrafish larval brain.
- All of the behavioral assay were performed with amyloid-beta concentrations up to 100fM, however, when the neurotransmitters were assayed it required 600fM to give any affect. The authors need to discuss this further and also comment on the physiological relevance of the amount of amyloid-beta that is being injected into the brain.
- I think the last section on the biomarkers and ROS assays should either be placed into supplementary data or more experiments performed to strengthen the conclusions on this section. I do not think the biomarkers analysed give a good indication of neurological dysfunction or neurodegeneration.
- The term neurodegeneration should be removed from the legend for figure 6.

Methods/Figure comments:

- What specific part of the brain was the amyloid-beta injected into?
- What areas of the brain is sectioned for analysis and what areas are presented in fig 3 and 6.
- All raw data on larval numbers for analysis should be included in supplementary data and the n=? should be in all figure legends.
- Figure 1D appears to have the incorrect information in legend. The percentages given correspond to random coils not alpha-helices.

Grammar/Readability:

- Sentence 43-47 is too long.
- Sentences 109-110, 196-200, 216-218 and 242-248 are out of place and do not flow with the paragraph.
- Sentences 179-181 should be re-written, specifically the term orthologue, not analogue, should be used. Also state the genes/proteins in the gamma-secretase complex.
- Sentence 212-213 should be re-written.

Overall this is a sound study on assessing the capability of beta-casein AuNPs to sequester amyloid-beta in-vitro and in the zebrafish larval brain. There is good evidence that the complex can ameliorate the behavioural pathology upon amyloid-beta injection but more evidence is required to provide support for the conclusions that ROS and neuronal dysfunction can also be ameliorated.

Reviewer #2 (Remarks to the Author):

The development of effective disease-modifying therapy to treat Alzheimer's disease is a true unmet need. The manuscript is describing the use of casein coated-gold nanoparticles to tackle the symptoms of amyloid-beta accumulation. The model that is used is zebrafish larvae with the injection of amyloid-beta into the brain. There are several issues to be discussed:

1. The authors discuss in the "Introduction" section the failure in clinical trials (even to a limited extent – please see my comments below). However, I assume that if those failed drugs would have been tried in the Zebrafish model that would have been successful. There is a clear need for control to show why the approach (protein-coated Au particles) is better for the development of anti-amyloid therapy.
2. Moreover, the mice model appears to be closer to human physiology and pathology. How the successful study in zebrafish does is enough for drug development?
3. The huge advantage of the zebrafish model is High-Throughput Screening. Clearly compared to mice... However, no screening was done here. No real optimization of the parameters of the casein coated-gold nanoparticles. The manuscript does not really take advantage of the uniqueness of the model.

Minor issues:

1. There many more withdrawals of clinical trials than those of Eli Lilly and Pfizer so the data is incomplete. On the other hand, the recent trial of Biogen appears to be successful.
2. Page numbers in reference #33 – should be 498-503

Reviewer #3 (Remarks to the Author):

In this report by Javed et al, the authors proposed the use of whey casein coated gold nanoparticles (β Cas-AuNPs) to remove A β 42 fibers (administered via cerebroventricular microinjection) from the brain of zebrafish larvae. The authors concluded that the β Cas-AuNPs reduced A β 42 neurotoxicity in zebrafish, using bioassays, optical and electron microscopy, and behavior monitoring. Overall the results and well presented but there are several points that not fully sustain the conclusions or that may require further discussion.

- the activity of β Cas over A β 42 is not clear, is it sequestration and inhibition, or a real chaperon activity? (text page 5) If it is the former it will infer a more catalytic or active role of the protein whereas the second, will mean that β Cas as chaperon will directly affect the secondary/tertiary structure of A β 42 into its native or stable structure. The authors need to expand the discussion in this point.
- the assumption that a cerebroventricular injection of A β in zebrafish larvae is a model for Alzheimer's disease (AD) is questionable, since AD is more complex than that, and here none of the histopathological hallmarks were presented or explored, and the very basic factor of old-age is not even discussed. Under this presumption it will mean that cerebroventricular injection of A β in any lab animal (mice, rat, rabbit, marmoset, dog, etc.) will immediately convert them in a model for AD. I strongly recommend to do not use this assumption.
- lines 108 and 109, it means that fibrillization of A β 42 stopped into the oligomer states?
- lines 109 and 110, β Cas was initially into a random coils state? It means that under this

unstructured state the protein will not have the chaperon-like activity, and can you discriminate if the α -helices were from β Cas or A β ?

- page 7. The DMD simulations were performed on 4 nm gold nanoparticles interacting with β Cas. Here it is assumed the formation of protein layer over the preformed metal core. But going to the Methods, the Au nanoparticles were produced directly in the presence of β Cas as a biotemplated or biodirected method. The simulations that should be performed were of the precursor Au³⁺ not the 4nm particle. The direct formation of Au nanoparticles in the presence of β Cas will cause changes in the structure, conformation and assembly of the protein, also addition of NaBH₄ might cause even more changes in the protein structure, since it is a strong reductant. Under these circumstances is not clear if the 3D structure of the β Cas was preserved.

- Figure 2. It will be necessary to have the simulations of bare AuNP with A β . To confirm or dismiss your predictions of formation of protein corona over Au, it is required additional TEM imaging (negative staining of proteins).

- Going back to the proposed AD model, the zebrafish used were only 5 days old, but this organism has a lifespan of 42 months, the age factor was not taken into account. A β accumulation is observed in elderly (65 years or older).

- The formation of A β plaques upon injection (Fig 3I) is not clear, and should be confirmed with additional analysis, like immunohistochemistry, or polarized light.

- For the A β toxicity and pathological symptoms section (Figure 5), no controls of pure AuNP were included. It seems that the effects are by Au not the β Cas protein.

Response Letter

We thank the reviewers for their insightful comments and endorsements of our manuscript. We have carefully studied these comments and incorporated them in the revised manuscript. In addition, we have carried out TEM, CD, histology, polarized light microscopy, in vitro viability, and in vivo cognition of adult zebrafish assays to accommodate the suggestions by the reviewers. All changes are shown in red in the main text and in the Supporting Information for easy viewing. We have also provided additional videos (Videos S4-7) to demonstrate administration of A β and casein AuNPs in zebrafish larvae and adults. Our point-by-point response is listed below for your assessment.

Reviewers' comments:

Reviewer #1 (Remarks to the Author):

The submitted manuscript by Javed and colleagues outlines a novel method of using gold nano-particles coated with beta-casein to sequester amyloid beta-42 peptides both in-vitro and in the brain of zebrafish larvae. The main conclusion of the manuscript is that the nano-particles may have broad use for eradication of toxic amyloid proteins in a range of human diseases. They provide behavioral data, ROS analysis and neuronal dysfunction biomarker assays to support their main conclusion. My expertise is in the zebrafish model and Alzheimer's disease therefore most of my comments will be focused on these aspects of the manuscript. Please note I have no particular expertise in molecular dynamics algorithms or simulations. Comments and queries:

- The Introduction mentions that both beta-casein and AuNPs both have chaperone like activity. Can the authors elaborate more on this activity and perhaps include more information in the introduction about this complex? Does this complex cross the BBB in humans?

Ans. We thank the reviewer for this insight. We have included an introduction to the chaperone-like activity of casein on p5 of the revised manuscript.

AuNPs do not possess inhibitory/chaperone or acceleratory/catalytic activity towards the amyloidosis, as clarified on p5. Ligation of β -casein on the surface of AuNPs, as presented in this manuscript, bestowed them with a chaperone-like capacity.

Since this is the first report exploring β Cas AuNPs of 5 nm size in vivo, there is no literature available for this complex to translocate across the human BBB. However, peptides-conjugated AuNPs have been widely employed for drug delivery, cancer imaging and their associated surface plasmon resonance (SPR) has been used to destruct tumour mass in vivo via photothermal radiation.¹ Peptides-conjugated AuNPs (10-20 nm) have been successfully implemented to target brain and neck tumours in mouse models, indicating the ability of protein-AuNP complexes in crossing the BBB.^{2,3} Furthermore, AuNPs of larger size than those studied in this manuscript have been used in different clinical trials for the treatment of head/brain and neck cancer in human, which also indicates the ability of peptide-AuNP complexes to translocate across the human BBB.⁴

- Further discussion is required on other amyloid diseases. There is still a huge debate on the role of “toxic” amyloid proteins in the AD, this should also be mentioned.

Ans. Thanks for this important suggestion. We have included three main forms of amyloid diseases as well as their origins of toxicity in the Introduction, on p4.

- I do not agree that amyloid beta is inducing a “Alzheimer's-like pathogenesis”. There is no discussion in the manuscript on how the behavioral pathology is similar to clinical symptoms observed in this disease. I think the authors should consider changing this term and also whether this actually is an AD-like zebrafish model.

*Ans. We agree with the reviewer that behavioural pathology observed with zebrafish larvae, i.e., reduced movement & frequency, is not what is always observed in the clinical symptoms of AD patients. Accordingly, we have replaced the 'Alzheimer's like pathogenesis' with 'A β toxicity' in the title and throughout the paper. However, A β did induce cognitive dysfunction in adult zebrafish (2 weeks post injection) in the additional experiments for the revision (**Figure 6**), which is more relevant to the clinical symptoms of AD patients and could be regarded as Alzheimer's-like symptoms. The advantages offered by the larvae include high-throughput analysis, a transparent and simple in vivo model, a vertebrate nervous system, similarities to human genome, and presence of genes and proteins that are orthologous to human diseases and, as studied in this manuscript, early development of the A β phenotype (3-5 days) elicited with very small sample volumes. This point has been included on p14-15.*

- The authors need to discuss the limitations of the model. This is a larval model and the fish are still developing so how does this relate to an aging disease in particular when looking at neurotransmitters in the zebrafish larval brain.

*Ans. Thanks for this insightful comment. As discussed in the previous question, the behavioural pathology in zebrafish larvae is not clinically relevant to AD patients. However, in additional experiments for the revision, adult zebrafish demonstrated cognitive impairment that is more relevant to clinical symptoms (**Figure 6**). The difference can be attributed to the fact that the nervous system in larva is not fully matured to adopt cognitive or memory function but developed enough⁵⁻⁷ to offer a quick vertebrate model system for A β toxicity. Although zebrafish lacks the advanced and complex cognitive behaviours as offered by rodents it can be used as a model to complement and enhance the understanding of disease and pharmacology of therapeutic modalities.⁸ We have added this to the Results and Discussion (p14-15) as well as in the Conclusion (p16).*

- All of the behavioral assay were performed with amyloid-beta concentrations up to 100fM, however, when the neurotransmitters were assayed it required 600fM to give any affect. The authors need to discuss this further and also comment on the physiological relevance of the amount of amyloid-beta that is being injected into the brain.

*Ans. Thanks for this question. According to literature, the A β_{42} concentration in the grey and white matter of the brain of AD patients is 1.3 and 0.25 nM/mg, respectively.⁹ Tg2576 mice models of AD present around 1600-1700 fM/mg of A β_{42} after developing the disease symptoms.¹⁰ However, the wet weight of whole zebrafish larva is ~ 1 mg and the brain is 4-5 times smaller than the whole larva.¹¹ Considering this physiological relevance of weight, A β_{42} was injected in zebrafish larvae over a concentration range of 0.07 to 1200 fM per larva and based on locomotor response (**Figure 3C**) 100 fM was selected for further experiments. However, A β of 100 fM did not produce any difference in neurotransmitter levels that are usually disturbed in severe cases of AD.¹² Therefore, a higher concentration of A β (600 fM) than the pathologically relevant concentration (100 fM) was used to observe possible fluctuations in neurotransmitters. As suggested by the reviewer in the next question, we have moved the neurotransmitters and ROS parts to the SI, and added this discussion to p2 of the SI.*

- I think the last section on the biomarkers and ROS assays should either be placed into supplementary data or more experiments performed to strengthen the conclusions on this section. I do not think the biomarkers analysed give a good indication of neurological dysfunction or neurodegeneration.

*Ans. We agree with the reviewer. As the concentration of A β required to disturb the neurotransmitters in zebrafish larvae was much higher than for the pathological concentration, it cannot be a direct indicator of neurological dysfunction. The figure has been moved to the SI and, in addition, synaptophysin loss was also imaged via immunostaining (**Figure S15E**).*

- The term neurodegeneration should be removed from the legend for figure 6.

Ans. The term has been removed as requested by the reviewer.

Methods/Figure comments:

- What specific part of the brain was the amyloid-beta injected into?

Ans. 50 nL of A β 42 was injected via cerebroventricular microinjection. The tip of the glass capillary needle was inserted in the ventricular space, across the dorsal soft skin tissue. The tip did not penetrate more than 0.1-0.3 mm across the center meeting point of left and right telencephalon. This information has been included in the Materials and Method section, on p21.

- What areas of the brain is sectioned for analysis and what areas are presented in fig 3 and 6.

*Ans. Whole larvae were fixed and sectioned for brain tissues. In particular, **Figs 3&6 (now Figs 3&5)** present the sagittal sections of the brain tissues. This information has been included in the legends of respective figures.*

- All raw data on larval numbers for analysis should be included in supplementary data and the n=? should be in all figure legends.

Ans. Thanks for the suggestion. We have included the raw excel data files for all figures. The numbers of repetitions and animals per group or per sample have also been included in the figure legends.

- Figure 1D appears to have the incorrect information in legend. The percentages given correspond to random coils not alpha-helices.

Ans. This oversight has been corrected.

Grammar/Readability:

- Sentence 43-47 is too long.

We have changed this to two sentences, on p2-3.

- Sentences 109-110, 196-200, 216-218 and 242-248 are out of place and do not flow with the paragraph.

We have rephrased sentences 109-110 on p7, deleted the redundant sentences 196-200 in original submission, moved sentences 216-218 to p15, deleted non-relevant information from sentences 242-248 and moved the description on the BBB up to p12.

- Sentences 179-181 should be re-written, specifically the term orthologue, not analogue, should be used. Also state the genes/proteins in the gamma-secretase complex.

We have made changes according to the reviewer's comments, on p11.

- Sentence 212-213 should be re-written.

We have revised this sentence on p15.

Overall this is a sound study on assessing the capability of beta-casein AuNPs to sequester amyloid-beta in-vitro and in the zebrafish larval brain. There is good evidence that the complex can ameliorate the behavioural pathology upon amyloid-beta injection but more evidence is required to provide support for the conclusions that ROS and neuronal dysfunction can also be ameliorated.

Ans. We thank the reviewer for endorsing our study with constructive and insight comments, which have significantly benefited the presentation and quality of the revised manuscript.

Reviewer #2 (Remarks to the Author):

The development of effective disease-modifying therapy to treat Alzheimer's disease is a true unmet need. The manuscript is describing the use of casein coated-gold nanoparticles to tackle the symptoms of amyloid-beta accumulation. The model that is used is zebrafish larvae with the injection of amyloid-beta into the brain. There are several issues to be discussed:

1. The authors discuss in the "Introduction" section the failure in clinical trials (even to a limited extent – please see my comments below). However, I assume that if those failed drugs would have been tried in the Zebrafish model that would have been successful. There is a clear need for control to show why the approach (protein-coated Au particles) is better for the development of anti-amyloid therapy.

*Ans. We thank the reviewer for this insight. We agree that successes in animal models like zebrafish, mice, rats or primates do not guarantee their success in human clinical trials. Drugs which pass phases 0 and 1 trials where human subjects are involved and pharmacokinetics and pharmacological efficacy are evaluated, can fail in phases 2-4 clinical trials where safety, efficacy, pharmaco and toxicokinetics of the drugs are evaluated on larger population sizes in multicentred trials. One example is Eli Lilly's Solanezumab that failed in phase 3 clinical trial where it didn't produce significant results shown in preclinical studies (animal models) and in phases 0-2 clinical trials (human subjects).¹³ However, use of simplified animal models can provide mechanistic knowledge on the pathology of the disease and pharmacology for the therapeutic modality. Zebrafish offers the advantages of high-throughput analysis, a transparent, simplified and yet full in-vivo model system, a vertebrate nervous system, similarities to human genome, presence of genes and proteins that are orthologous to human diseases and, as studied in this manuscript, early development of A β phenotype (3-5 days) administered with very small sample volumes. This point has been included on p14&15. Recently, invertebrates *Caenorhabditis elegans* nematodes¹⁴⁻¹⁸ and *drosophila*^{19,20} have also been employed as simple in-vivo models to study the neurodegeneration and behavioural pathologies of protein aggregation diseases.*

At this stage, there is not FDA-approved drug available for Alzheimer's disease with particular mechanisms of sequestration and clearance of A β burden from the brain that could be compared with β Cas AuNPs as control. However, β Cas, buffer alone, untreated control, A β alone and citrate AuNPs have been included as controls for the revision.

2. Moreover, the mice model appears to be closer to human physiology and pathology. How the successful study in zebrafish does is enough for drug development?

Ans. We agree with the reviewer that mouse models are complex and closer to represent human physiology and pathology. One of the objectives of this study is to assess how a NP-based chaperone approach can work against A β amyloidosis in vivo. Although zebrafish is not as advanced as rodents it offers a genetically manipulatable visual model to study the detailed pathology of diseases due to expressing orthologue genes and proteins of human diseases.^{8,21} In this study, zebrafish larvae is developed and used as a facile in vivo model to study A β fibrillization, toxicity, behavioural pathology, neurodegeneration, biodistribution and nano-chaperone activity of β Cas AuNPs. We have added this content to the Results and Discussion (p14-15) as well as in the Conclusion (p16).

*We have performed additional experiments using adult zebrafish as a mature and more complex in vivo model²² to study the clinically relevant toxicity of A β , i.e., behavioural and cognitive dysfunction, and subsequent mitigation with β Cas AuNPs (**Figure 6**).*

Studies with zebrafish for neurodegeneration diseases can provide an initial in vivo and high-throughput platform to understand the pathology of neurodegeneration, associated behavioural pathology and pharmacology of drugs, as presented in this manuscript.

3. The huge advantage of the zebrafish model is High-Throughput Screening. Clearly compared to mice... However, no screening was done here. No real optimization of the

parameters of the casein coated-gold nanoparticles. The manuscript does not really take advantage of the uniqueness of the model.

Ans. Thanks for this insight. We agree with the reviewer that a major advantage of zebrafish larvae, in addition to their transparency, is high-throughput screening of large libraries of medicinal compounds to find the HIT leads. In this study, screening of the toxic dose for A β was performed via high throughput (Figure 3C). Due to the large number and variety of assays performed for the study, we used only one leading sample of β Cas AuNPs, so no therapeutic screening was required. However, β Cas AuNPs were studied (high throughput) using a large number of larvae (n=20 per group and 3 groups per sample) in 96 well plates (Figure S17). Regarding the optimization of β Cas AuNPs, the method, reaction conditions and β Cas to Au ratio were optimized to obtain the small (5 nm) monodisperse nanoparticles for efficient BBB translocation. This information has been included in the Discussion (p6). By changing the ratio of the ingredients or performing the reaction at 4 °C resulted in larger sized polydisperse particles while increasing the reaction temperature resulted in a decrease of the random coils or increase of the secondary structure of β Cas corona in β Cas AuNPs. The 5 nm β Cas AuNPs were then fully characterized for their physical parameters and in-vitro chaperone interaction with A β (Figures I&SI, Table SI). Additional SPR and fluorescence characterizations of β Cas AuNPs and neutral red-conjugated NR- β Cas AuNPs, before and after their interactions with A β monomers and oligomers have also been performed (Figure S2).

The advantages offered by the larvae include a high throughput analysis, transparent and simplified yet full in-vivo model system, vertebrate nervous system, similarity to human genome, and presence of genes and proteins that are orthologous to human diseases and, as studied in this manuscript, early development of A β phenotype (3-5 days) elicited with small sample volumes. Furthermore, additional experiments with adult zebrafish revealed the clinical relevance of the behavioural pathology and memory/cognitive dysfunction (Figure 6).

Minor issues:

1. There many more withdrawals of clinical trials than those of Eli Lilly and Pfizer so the data is incomplete. On the other hand, the recent trial of Biogen appears to be successful.

Ans. Eli Lilly's solanezumab and Pfizer's ACC-001 and QS-21 failed in phase 3 clinical trials. Biogen Inc and partner Eisai Co Ltd recently announced (21 March, 2019) the withdrawal of aducanumab in phase 3 trials. All these therapeutic modalities were based on monoclonal antibodies. This is a major setback for the quest for anti-Alzheimer's drug, further reinforcing the urgency of exploring alternative strategies against AD.^{23,24}

2. Page numbers in reference #33 – should be 498-503.

Ans. We have corrected this oversight (now reference 43).

Reviewer #3 (Remarks to the Author):

In this report by Javed et al, the authors proposed the use of whey casein coated gold nanoparticles (β Cas-AuNPs) to remove A β 42 fibers (administered via cerebroventricular microinjection) from the brain of zebrafish larvae. The authors concluded that the β Cas-AuNPs reduced A β 42 neurotoxicity in zebrafish, using bioassays, optical and electron microscopy, and behavior monitoring. Overall the results and well presented but there are several points that not fully sustain the conclusions or that may require further discussion.

- the activity of β Cas over A β 42 is not clear, is it sequestration and inhibition, or a real chaperon activity? (text page 5) If it is the former it will infer a more catalytic or active role of the protein whereas the second, will mean that β Cas as chaperon will directly affect the secondary/tertiary structure of A β 42 into its native or stable structure. The authors need to expand the discussion in this point.

Ans. We thank the reviewer for this discussion. β Cas has chaperon-like activity, particularly towards amyloid proteins. Specifically, α_s and β -casein possess chaperone like activity, similar to small heat-shock proteins (sHSP) and extracellular clusterin. This activity arises from (1) a lack of tertiary structure and solvent exposed hydrophobicity with well separated hydrophilic regions, (2) existence as heterogeneous oligomers, (3) dynamics and malleable protein regions, (4) ability to bind with a wide range of partially folded proteins preventing their aggregation.²⁵ One reason that attributes to these properties is the presence of high percentage of proline residues, i.e., 18% in case of β -casein, and no disulphide bonds that provide them with an open and flexible conformation.²⁶ The chaperone behaviour of α_s and β -casein shields the amyloidogenic regions and naturally prevents the amyloidosis of α_{s2} and κ -casein²⁷ in mammary glands or milk while inhibiting the amyloidosis of insulin and amyloid- β_{40} in vitro.^{28,29} This content has been included to p5.

We agree with the reviewer that sequestration of amyloid protein around NPs can catalyse the fibrillization process due to highly localized concentration of amyloid protein in the 'halo' of the NPs. However, it also depends upon the surface chemistry of the NPs. Surface ligands with inhibitory or catalytic capacity can assign the NPs with the same activity, as observed in this study that citrate AuNPs did not inhibit $A\beta$ fibrillization (**Figure 5A**) as opposed to β Cas AuNPs. Citrate AuNPs have been reported to possess a catalytic behaviour towards fibrillization.^{30,31} This information has been included in the Introduction (p5) as well as in Results and Discussion (p14). As β Cas physically interacted and shielded the amyloidogenic segments to prevent $A\beta$ self-assembly, such interaction resulted in inhibition of $A\beta$ fibrillization and directly influenced the secondary/tertiary structure of the peptide (**Figure 1D**). This has been included in the Results and Discussion (p8).

- the assumption that a cerebroventricular injection of $A\beta$ in zebrafish larvae is a model for Alzheimer's disease (AD) is questionable, since AD is more complex than that, and here none of the histopathological hallmarks were presented or explored, and the very basic factor of old-age is not even discussed. Under this presumption it will mean that cerebroventricular injection of $A\beta$ in any lab animal (mice, rat, rabbit, marmoset, dog, etc.) will immediately convert them in a model for AD. I strongly recommend to do not use this assumption.

Ans. Thanks for this important comment. Yes, we agree with the reviewer that injection of $A\beta$ to any animal cannot directly convert it to a full AD model. We have replaced the term of AD model with $A\beta$ toxicity in the title and throughout the paper to reflect the observations of $A\beta$ toxicity and behavioural pathology. However, inducing neurodegeneration and behavioural pathology with cerebroventricular injections of either monomeric, oligomeric and even full-length fibrils of $A\beta$ or α -synuclein is evident from the literature,³²⁻³⁹ even though these models cannot be regarded as full AD or PD models as they do not include the factor of old-age and other complex in-vivo pathologies.

To take into consideration of the age factor, we have performed additional experiments with adult zebrafish (10 months). Adult zebrafish offer fully developed the brain anatomy and cognitive behaviour.²² Thus, it presented with clinically relevant symptoms of cognitive dysfunction upon injection with $A\beta$ (**Figure 6**). β Cas AuNPs mitigated $A\beta$ toxicity in adult zebrafish (**Figure 6**) that further supported the in-vivo efficacy of chaperone-NPs against amyloidosis. In the context of histological hallmarks, we observed $A\beta$ deposition in the brain tissue via fluorescence, immunohistochemistry (IHC) and polarized light microscopy (**Figure 5C,D**).

- lines 108 and 109, it means that fibrillization of $A\beta_{42}$ stopped into the oligomer states?

Ans. β Cas AuNPs sequestered $A\beta$ on their surfaces, as observed by the $A\beta$ corona under TEM (**Figure 1F inset**). CD results suggested a transition of the secondary structure of β Cas AuNPs from random coils to α -helices in the $A\beta$ - β Cas AuNPs complex (**Figure 1D**). As CD described the dominant protein conformation in the solution and together with the fact that $A\beta$ corona buried the β Cas conformation, it is reasonable to assume that the α -helix conformation observed in $A\beta$ - β Cas AuNPs was predominantly from the $A\beta$ corona. This indicates the possibility of $A\beta$ in the form of either oligomeric state or the conformation of $A\beta$ monomers was transitioned to α -helices in the $A\beta$ - β Cas AuNPs complex. This aspect has been included in the Discussion (p8). Furthermore, as discussed in the next question,

additional experiments (**Figure S2**) demonstrated the ability of β Cas AuNPs to preferably sequester $A\beta$ in the oligomeric state.

- lines 109 and 110, β Cas was initially into a random coils state? It means that under this unstructured state the protein will not have the chaperon-like activity, and can you discriminate if the α -helices were from β Cas or $A\beta$?

Ans. β Cas possesses the random coil conformation in its native state. As discussed on p5 that open, dynamic and flexible or unstructured conformation together with solvent-exposed and well-spaced hydrophobic (major) and hydrophilic (minor) zones are the responsible factors for the chaperone activity of α - and β -casein.^{25,27-29} TEM images showed the outer corona of $A\beta$ (**Figure 1F inset**), thus β Cas on AuNPs should be buried under $A\beta$. This indicates the α -helices conformations were predominantly from $A\beta$.

We performed additional experiments by incubating β Cas AuNPs with $A\beta$ monomers and oligomers for 3 h, separating the NPs via centrifugal washing and analysing via UV-SPR, fluorescence, CD and TEM to investigate the relative affinity of β Cas AuNPs for $A\beta$ oligomers vs monomers (**Figure S2**). The results suggested that β Cas AuNPs had a greater affinity to bind with $A\beta$ oligomers than monomers. $A\beta$ oligomers were adsorbed on the surfaces of β Cas AuNPs, thus the α -helices in $A\beta$ - β Cas AuNPs were from the $A\beta$ corona. The ability of β Cas AuNPs to preferentially bind with $A\beta$ was also evident in the BCA quantification assay (**Figure 1L**). This content has been added to the Discussion, on p8.

- page 7. The DMD simulations were performed on 4 nm gold nanoparticles interacting with β Cas. Here it is assumed the formation of protein layer over the preformed metal core. But going to the Methods, the Au nanoparticles were produced directly in the presence of β Cas as a biotemplated or biodirected method. The simulations that should be performed were of the precursor Au^{3+} not the 4nm particle. The direct formation of Au nanoparticles in the presence of β Cas will cause changes in the structure, conformation and assembly of the protein, also addition of $NaBH_4$ might cause even more changes in the protein structure, since it is a strong reductant. Under these circumstances is not clear if the 3D structure of the β Cas was preserved.

Ans. As pointed out by the reviewer, β Cas was used as a surface capping agent while $NaBH_4$ was used as a reducing agent. One reason for using the biotemplated synthetic strategy was to possibly preserve the flexible and dynamic conformation of β Cas that is required for its chaperone activity. Addition of $NaBH_4$ reduced Au^{3+} to Au atoms that started aggregating. However, β Cas immediately capped the aggregation of Au resulting in 5 nm β Cas AuNPs. Furthermore, β Cas possesses one cysteine residue and no disulphide bridge, therefore the reducing activity of $NaBH_4$ was consumed in the reduction of Au^{3+} , while β Cas remained unaffected from the reducing action of $NaBH_4$. Hence, β Cas adopted its thermodynamically stable conformations on the AuNP surface and the structure of the β Cas corona remained independent of the synthesis process of AuNPs. Since our focus was to study the capping of β Cas AuNPs and the subsequent ability of β Cas corona to bind with $A\beta$ we therefore modelled only the binding of β Cas with a pre-formed AuNP also in consideration of the computational cost. We have added this to the Supplementary Methods in the SI.

- Figure 2. It will be necessary to have the simulations of bare AuNP with $A\beta$. To confirm or dismiss your predictions of formation of protein corona over Au, it is required additional TEM imaging (negative staining of proteins).

Ans. Following the reviewer's suggestion, we performed a control simulation of $A\beta$ binding with a bare AuNP. The binding probability of each residue with the AuNP and representative binding structures were added as **Figure S6** in the SI.

By approximating the protein-AuNP binding affinity with the energy difference between the complex and isolated components from DMD simulations, we found that binding of β Cas with AuNPs was significantly stronger than with $A\beta$, i.e., $\Delta G \sim -194$ kcal/mol (**Table S2**). Hence, we did not expect $A\beta$ would replace β Cas in the corona. We have added this to the Discussion, on p10-11.

As suggested by the reviewer, additional TEM imaging (negative staining) of A β corona in A β - β Cas AuNPs has been performed (**Figure 1F inset, Figure S2C**), showing the binding of A β to the nanoparticles.

- Going back to the proposed AD model, the zebrafish used were only 5 days old, but this organism has a lifespan of 42 months, the age factor was not taken into account. A β accumulation is observed in elderly (65 years or older).

*Ans. Thanks for suggesting this important factor of age. We have performed additional experiments with adult zebrafish (10 months) to study the influence of age. As adult zebrafish possess more mature cognitive function,²² A β microinjection revealed clinically relevant behavioural pathology of cognitive dysfunction (**Figure 6**). Furthermore, β Cas AuNPs mitigated the A β toxicity in adult zebrafish (**Figure 6**) that supports the in vivo efficacy of chaperone NPs against amyloidosis.*

- The formation of A β plaques upon injection (Fig 3I) is not clear, and should be confirmed with additional analysis, like immunohistochemistry, or polarized light.

*Ans. Thanks for suggesting these hallmark characterisations for A β plaques. We have performed immunohistochemistry and polarized light microscopy to detect the presence of fibrillized A β in the brain tissue (sagittal plane) of zebrafish larvae (**Figure 5C,D**). Immunohistochemistry for A β has also been performed on adult zebrafish brain sections (**Figure 6C**).*

- For the A β toxicity and pathological symptoms section (Figure 5), no controls of pure AuNP were included. It seems that the effects are by Au not the β Cas protein.

Ans. Thanks for suggesting this control. AuNPs do not possess native inhibitory/chaperone or acceleratory/ catalytic activity towards amyloidosis. AuNPs only exist in the presence of surface stabilizing agents like PEG, citrate or proteins. The ability of AuNPs to act as nano-chaperones or nano-catalysts is controlled by the nature of their surface ligands.

*In order to rule out the possible effect of AuNPs alone, additional experiments with citrate AuNPs have been performed. Citrate is the smallest and stable surface ligands for AuNPs and is considered as closely relevant to bare AuNPs. As expected, Citrate AuNPs were not able to rescue the A β toxicity in zebrafish larvae (**Figure 5A**), unlike β Cas AuNPs.*

References

- 1 Pedrosa, P., Vinhas, R., Fernandes, A. & Baptista, P. Gold nanotheranostics: proof-of-concept or clinical tool? *Nanomaterials* **5**, 1853-1879 (2015).
- 2 Hainfeld, J. F., Smilowitz, H. M., O'connor, M. J., Dilmanian, F. A. & Slatkin, D. N. Gold nanoparticle imaging and radiotherapy of brain tumors in mice. *Nanomedicine* **8**, 1601-1609 (2013).
- 3 Meola, A., Rao, J., Chaudhary, N., Sharma, M. & Chang, S. D. Gold nanoparticles for brain tumor imaging: a systematic review. *Front. Neurol.* **9**, 328 (2018).
- 4 Singh, P. *et al.* Gold nanoparticles in diagnostics and therapeutics for human cancer. *Int. J. Mol. Sci.* **19**, 1979 (2018).
- 5 Raible, D. W. & Kruse, G. J. Organization of the lateral line system in embryonic zebrafish. *J. Comp. Neurol.* **421**, 189-198 (2000).
- 6 Myers, P. Z. Spinal motoneurons of the larval zebrafish. *J. Comp. Neurol.* **236**, 555-561 (1985).
- 7 Bernhardt, R. R., Chitnis, A. B., Lindamer, L. & Kuwada, J. Y. Identification of spinal neurons in the embryonic and larval zebrafish. *J. Comp. Neurol.* **302**, 603-616 (1990).
- 8 Newman, M., Ebrahimie, E. & Lardelli, M. Using the zebrafish model for Alzheimer's disease research. *Front. Genet.* **5**, 189 (2014).

- 9 Roher, A. E. *et al.* Amyloid beta peptides in human plasma and tissues and their significance for Alzheimer's disease. *Alzheimer's & Dementia* **5**, 18-29 (2009).
- 10 Pacheco-Quinto, J. *et al.* Hyperhomocysteinemic Alzheimer's mouse model of amyloidosis shows increased brain amyloid β peptide levels. *Neurobiol. Dis.* **22**, 651-656 (2006).
- 11 Avella, M. A. *et al.* Lactobacillus rhamnosus accelerates zebrafish backbone calcification and gonadal differentiation through effects on the GnRH and IGF systems. *PLoS One* **7**, e45572 (2012).
- 12 Sáez-Valero, J., Sberna, G., McLean, C. A. & Small, D. H. Molecular isoform distribution and glycosylation of acetylcholinesterase are altered in brain and cerebrospinal fluid of patients with Alzheimer's disease. *J. Neurochem.* **72**, 1600-1608 (1999).
- 13 Honig, L. S. *et al.* Trial of solanezumab for mild dementia due to Alzheimer's disease. *N. Engl. J. Med.* **378**, 321-330 (2018).
- 14 Cornaglia, M. *et al.* in *Conference Proceedings*. 689-691.
- 15 Cornaglia, M. *et al.* Automated longitudinal monitoring of in vivo protein aggregation in neurodegenerative disease *C. elegans* models. *Mol. Neurodegener.* **11**, 17 (2016).
- 16 Dostal, V. & Link, C. Assaying β -amyloid Toxicity using a Transgenic *C. elegans* Model. *JoVE*. 44. *J. Vis. Exp.* **44**, e2252 (2010).
- 17 Link, C. D. *et al.* Visualization of fibrillar amyloid deposits in living, transgenic *Caenorhabditis elegans* animals using the sensitive amyloid dye, X-34. *Neurobiol. Aging* **22**, 217-226 (2001).
- 18 Dosanjh, L. E., Brown, M. K., Rao, G., Link, C. D. & Luo, Y. Behavioral phenotyping of a transgenic *Caenorhabditis elegans* expressing neuronal amyloid- β . *J. Alzheimers Dis.* **19**, 681-690 (2010).
- 19 Rival, T. *et al.* Fenton chemistry and oxidative stress mediate the toxicity of the β -amyloid peptide in a *Drosophila* model of Alzheimer's disease. *Eur. J. Neurosci.* **29**, 1335-1347 (2009).
- 20 Auluck, P. K., Chan, H. E., Trojanowski, J. Q., Lee, V. M.-Y. & Bonini, N. M. Chaperone suppression of α -synuclein toxicity in a *Drosophila* model for Parkinson's disease. *Science* **295**, 865-868 (2002).
- 21 Yan, C. *et al.* Visualizing Engrafted Human Cancer and Therapy Responses in Immunodeficient Zebrafish. *Cell* (2019).
- 22 Lau, B. Y., Mathur, P., Gould, G. G. & Guo, S. Identification of a brain center whose activity discriminates a choice behavior in zebrafish. *Proc. Natl. Acad. Sci. U.S.A.* **108**, 2581-2586 (2011).
- 23 Steenhuysen, J. *Biogen scraps two Alzheimer drug trials, wipes \$18 billion from market value*, <<https://www.reuters.com/article/us-biogen-alzheimers/biogen-scraps-two-alzheimer-drug-trials-wipes-18-billion-from-market-value-idUSKCN1R213G>> (2019).
- 24 Feuerstein, A. *Biogen halts studies of closely watched Alzheimer's drug, a blow to hopes for new treatment*, <<https://www.statnews.com/2019/03/21/biogen-eisai-alzheimer-trial-stopped/>> (2019).
- 25 Thorn, D. C., Ecroyd, H. & Carver, J. A. The two-faced nature of milk casein proteins: amyloid fibril formation and chaperone-like activity. *Aust. J. Dairy Technol.* **64**, 34-40 (2009).
- 26 Guha, S., Manna, T. K., Das, K. P. & Bhattacharyya, B. Chaperone-like activity of tubulin. *J. Biol. Chem.* **273**, 30077-30080 (1998).
- 27 Thorn, D. C. *et al.* Amyloid fibril formation by bovine milk κ -casein and its inhibition by the molecular chaperones α S-and β -casein. *Biochemistry* **44**, 17027-17036 (2005).
- 28 Librizzi, F., Carrotta, R., Spigolon, D., Bulone, D. & San Biagio, P. L. α -Casein inhibits insulin amyloid formation by preventing the onset of secondary nucleation processes. *J. Phys. Chem. Lett.* **5**, 3043-3048 (2014).
- 29 Carrotta, R. *et al.* Inhibiting effect of α s1-casein on A β 1-40 fibrillogenesis. *BBA-Gen. Subjects* **1820**, 124-132 (2012).
- 30 Javed, I. *et al.* Probing the aggregation and immune response of human islet amyloid polypeptide with ligand-stabilized gold nanoparticles. *ACS Appl. Mater. Interfaces* **11**, 10462-10471 (2019).

- 31 Gladytz, A., Abel, B. & Risselada, H. J. Gold-Induced Fibril Growth: The Mechanism of Surface-Facilitated Amyloid Aggregation. *Angew. Chem. Int. Ed.* **55**, 11242-11246 (2016).
- 32 Lu, P. *et al.* Silibinin prevents amyloid β peptide-induced memory impairment and oxidative stress in mice. *Br. J. Pharmacol.* **157**, 1270-1277 (2009).
- 33 Cleary, J. P. *et al.* Natural oligomers of the amyloid- β protein specifically disrupt cognitive function. *Nat. Neurosci.* **8**, 79-84 (2005).
- 34 Ammassari-Teule, M., Middei, S., Passino, E. & Restivo, L. Enhanced procedural learning following beta-amyloid protein (1-42) infusion in the rat. *Neuroreport* **13**, 1679-1682 (2002).
- 35 Nakamura, S., Murayama, N., Noshita, T., Annoura, H. & Ohno, T. Progressive brain dysfunction following intracerebroventricular infusion of beta1-42-amyloid peptide. *Brain Res.* **912**, 128-136 (2001).
- 36 Maurice, T., Lockhart, B. P. & Privat, A. Amnesia induced in mice by centrally administered β -amyloid peptides involves cholinergic dysfunction. *Brain Res.* **706**, 181-193 (1996).
- 37 Stepanichev, M. Y., Moiseeva, Y. V., Lazareva, N., Onufriev, M. & Gulyaeva, N. Single intracerebroventricular administration of amyloid-beta (25-35) peptide induces impairment in short-term rather than long-term memory in rats. *Brain Res. Bull.* **61**, 197-205 (2003).
- 38 Christensen, R., Marcussen, A. B., Wörtwein, G., Knudsen, G. & Aznar, S. A β (1-42) injection causes memory impairment, lowered cortical and serum BDNF levels, and decreased hippocampal 5-HT_{2A} levels. *Exp. Neurol.* **210**, 164-171 (2008).
- 39 Winner, B. *et al.* In vivo demonstration that α -synuclein oligomers are toxic. *Proc. Natl. Acad. Sci. U.S.A.* **108**, 4194-4199 (2011).

REVIEWERS' COMMENTS:

Reviewer #1 (Remarks to the Author):

The authors have adequately addressed my comments and queries from the original submission.

Reviewer #2 (Remarks to the Author):

The authors had fully addressed all my concerns. I have no further queries.

Reviewer #4 (Remarks to the Author):

Although I do not buy amyloid toxicity--I think this is a major finding. Au-casein particle disruption of amyloid opens a new avenue. The authors should discuss the work suggesting amyloid is a response to disease that is much like inflammation. I think that is what reviewer #1 is asking. That alternative does not diminish the finding.

The authors should discuss why ALL animal models suffer from being created and thus treatable by removal of the insult. They are not real biology. Zebra fish, worms, mice are all created. They are all unlikely to predict human response directly. The value of the study is mainly that the Au beads can remove Abeta---that is great in itself.

We thank the reviewers for their strong endorsements of our manuscript. Our point-by-point response is provided below.

Reviewer #1 (Remarks to the Author):

The authors have adequately addressed my comments and queries from the original submission.

Reviewer #2 (Remarks to the Author):

The authors had fully addressed all my concerns. I have no further queries.

Reviewer #4 (Remarks to the Author):

Although I do not buy amyloid toxicity--I think this is a major finding. Au-casein particle disruption of amyloid opens a new avenue. The authors should discuss the work suggesting amyloid is a response to disease that is much like inflammation. I think that is what reviewer #1 is asking. That alternative does not diminish the finding.

The authors should discuss why ALL animal models suffer from being created and thus treatable by removal of the insult. They are not real biology. Zebra fish, worms, mice are all created. They are all unlikely to predict human response directly. The value of the study is mainly that the Au beads can remove Abeta---that is great in itself.

Yes we agree with the reviewer that the causes of Alzheimer's are complex, and the existing literature implicates both amyloidogenesis and inflammation, among others, as plausible reasons for the pathologies observed in Alzheimer's patients. We have acknowledged the role of inflammation in the Introduction on p3, and cited relevant papers (refs 6-9).